# CRISPR-Cas9 cytidine and adenosine base editing of splice-sites mediates highly-efficient disruption of proteins in primary and immortalized cells

Mitchell G. Kluesner [1,2,3,4,7], Walker S. Lahr[1,2,3,4,7], Cara-lin Lonetree[1,2,3,4], Branden A. Smeester [1,2,3,4], Xiaohong Qiu[1,2,3,4], Nicholas J. Slipek [1,2,3,4], Patricia N. Claudio Vázquez[1,2,3,4,5], Samuel P. Pitzen [2,5], Emily J. Pomeroy [1,2,3,4], Madison J. Vignes[6], Samantha C. Lee [5,6], Samuel P. Bingea[1,2,3,4], Aneesha A. Andrew[5,6], Beau R. Webber [1,2,3,4,8✉] & Branden S. Moriarity[1,2,3,4,8✉]

CRISPR-Cas9 cytidine and adenosine base editors (CBEs and ABEs) can disrupt genes without introducing double-stranded breaks by inactivating splice sites (BE-splice) or by introducing premature stop (pmSTOP) codons. However, no in-depth comparison of these methods or a modular tool for designing BE-splice sgRNAs exists. To address these needs, we develop SpliceR (http://z.umn.edu/spliceR) to design and rank BE-splice sgRNAs for any Ensembl annotated genome, and compared disruption approaches in T cells using a screen against the TCR-CD3 MHC Class I immune synapse. Among the targeted genes, we find that targeting splice-donors is the most reliable disruption method, followed by targeting splice-acceptors, and introducing pmSTOPs. Further, the CBE BE4 is more effective for disruption than the ABE ABE7.10, however this disparity is eliminated by employing ABE8e. Collectively, we demonstrate a robust method for gene disruption, accompanied by a modular design tool that is of use to basic and translational researchers alike.

[1] Department of Pediatrics, University of Minnesota, Minneapolis, MN, USA. [2] Masonic Cancer Center, University of Minnesota, Minneapolis, MN, USA. [3] Center for Genome Engineering, University of Minnesota, Minneapolis, MN, USA. [4] Stem Cell Institute, University of Minnesota, Minneapolis, MN, USA. [5] Department of Genetics, Cell Biology, and Development, University of Minnesota, Minneapolis, MN, USA. [6] College of Biological Sciences, University of Minnesota, Minneapolis, MN, USA. [7] These authors contributed equally: Mitchell G. Kluesner, Walker S. Lahr. [8] These authors jointly supervised this work: Beau R. Webber, Branden S. Moriarity. ✉email: webb0178@umn.edu; mori0164@umn.edu

Clustered Regularly Interspaced Short Palindromic Repeats (CRISPR) systems and their CRISPR associated proteins (Cas proteins) have allowed for an unprecedented ability to manipulate the genome[1–5]. Key amongst the applications of these systems is their use in gene editing for targeted gene knockout, knockin, and modification[6]. These applications are of particular interest in the field of cellular immunotherapies, where the multiplexed disruption of genes involved in alloreactivity (e.g., *TRAC, TRBC,* and *B2M*) and in immunosuppression (e.g., *PDCD1, CTLA4, TGFBR2,* and *CISH*) in tandem with the knockin of chimeric antigen receptors (CARs) specific to tumor antigens yields promise in the development of efficacious and safe therapies to recalcitrant malignancies[7–10]. In the most commonly used form of CRISPR, the Cas9 nuclease from *Streptococcus pyogenes* (hereafter referred to as Cas9) is paired with a single-guide RNA (sgRNA) to induce a double-stranded break (DSB) at a specific DNA site directed by the programmable complementarity of the 20-nt sgRNA protospacer[5]. While CRISPR-Cas9 nucleases work exceptionally well for single gene editing, multiple concerns have emerged surrounding DSB induction, including large scale genomic rearrangements[11,12], loss of heterozygosity[13], and selection of cells with p53 alterations[14], leading to suboptimal efficiency and the potential of oncogenesis in cell based therapies. Concerns arising from DSBs are exacerbated in a multiplex setting, where multiple genes are targeted simultaneously[7,15]. However, as more is understood about the complex genetic circuitry involved in cancer immunosurveillance[16], and enthusiasm increases for generating 'off the shelf' CAR T cells[1–5,7,15], it is of increased interest to edit genes in a multiplex setting[8].

An alternative tool to edit genes without causing DSBs are CRISPR-Cas9 base editors. Base editors are a class of gene editing enzymes that consist of a Cas9 nickase fused to a nucleotide deaminase domain[17–19]. In principle, base editors localize to a target region in the genome guided by a sgRNA. Once bound, the Cas9 complexes displaces the nonbound strand, forming a ssDNA R-loop. The R-loop is rendered accessible to the tethered deaminase domain, whereby cytidine deaminase base editors (CBEs, C:G-to-T:A) deaminate C-to-U, which base pairs like T, and adenosine deaminase base editors (ABEs, A:T-to-G:C) deaminate A-to-I, which base pairs like G. Concurrent nicking of the unedited strand by the core Cas9 nickase then stimulates DNA repair to use the newly deaminated base as a template for DNA polymerization, thereby preserving the edit in both strands of the DNA.

Previously, our group established a platform for the multiplex engineering of human lymphocytes using CRISPR-Cas9 base editors. This approach achieves high-efficiency multiplex editing without DSBs and their associated complications, such as chromosomal translocations and hindered cell proliferation[15]. In that work, we used CBEs to disrupt genes by mutating CAG, CGA, and TGG codons to introduce premature stop codons (pmSTOPs, previously termed iSTOP or CRISPR-STOP)[20,21], or by mutating the conserved splice-site motifs to disrupt RNA splicing. To date, there has been substantial interest in using both CBEs and ABEs to modulate splicing, which has predominantly focused on using targeted skipping of exons bearing pathogenic mutations[22,23] and inducing functional alternative splicing patterns[24]. However, substantial evidence supports the utility of splice-site targeting as a method for functional gene knockout as opposed to strictly modulating alternative splicing (here, distinguished as BE-splice). Despite the array of reports demonstrating the various ways base editors can be used to disrupt genes and modulate splicing, here we address three main gaps in the field, namely (1) a modular tool for designing both CBE and ABE sgRNAs to target both splice donor (SD) and splice acceptor (SA) sites, (2) a head-to-head comparison of the methods of base editing-mediated gene

disruption encompassing ABEs, CBEs, BE-splice methods, and pmSTOP induction, and (3) an investigation of the characteristics of successful BE-splice guides.

Here we present an easy-to-use web tool, SpliceR (http://z.umn.edu/spliceR), for the design of base editing sgRNAs to target splice-sites of any Ensembl annotated metazoan genome. We demonstrate the robustness of BE-splice in a focused sgRNA screen targeting the proteins that make up the heterooctameric T-cell receptor-CD3 (TCR-CD3) complex and MHC Class I immune synapse. Among the genes in our screen, we find that for gene editing and protein disruption (1) CBEs were more reliable than seventh generation ABEs, and comparable to eighth generation CBEs, (2) among both CBEs and ABEs, targeting splice donors tended to produce more reliable disruption than targeting splice acceptors, and (3) targeting splice-donor sites produced more robust disruption than pmSTOPs. Collectively, we describe a robust method and program for the design of sgRNAs for gene disruption via the base editing of splice sites.

## Results

**Development of SpliceR Algorithm.** In previous work, we and others established the ability to target splice sites with base editing for both modulating splicing patterns and disrupting proteins[15,22,24]. While programs exist for designing splice acceptor targeting guides with a limited number of PAMs, there is not a comprehensive program for designing guides that target both splice donors and splice-acceptors sites with CBEs or ABEs (Fig. 1a, b) that is compatible with any PAM identity or Ensembl annotated species. To meet these needs, we developed the program SpliceR for the design of BE-splice sgRNAs. SpliceR employs an interactive user interface to specify the parameters for BE-splice sgRNA design, such as Ensembl transcript ID, PAM identity, base editor type, and species of interest (Supplementary Fig. 1a). SpliceR communicates directly with Ensembl.org to query genetic information for sgRNA design and scoring.

To assess the practicality of the BE-splice approach, we used SpliceR to identify sgRNAs for every protein coding gene in the human genome. When restricting the analysis to genes that undergo splicing (i.e., spliced genes), we found that 99.85% of transcripts and 99.68% of spliced protein coding genes, which comprise 94.23% of all protein coding genes, are expected to be targetable (Fig. 1c). Furthermore, 50% of spliced genes are expected to have 62 sgRNAs or more, with the earliest sgRNA within the first 11.13% of the transcript (Fig. 1d, e). When broken down by targeting splice donors or splice acceptors, splice donors tend to have more sgRNAs (Supplementary Fig. 2), while also having the first sgRNA appear earlier in the transcript (Supplementary Fig. 3). These predictions suggested that the BE-splice approach is a robust method for targeting most genes with CBEs and-or ABEs.

**Screening BE-splice guides against the TCR-CD3 MHC class I immune synapse.** Despite the number of publications that have used base editors to disrupt or modulate genes by targeting splice sites with CBEs and ABEs or introducing pmSTOPs with CBEs[15,20,21,23–27], to our knowledge there is no comprehensive, direct comparison among these methods. Therefore, we sought to compare these methods directly and assess the predictions made from SpliceR in a medium-throughput screen. We saw the TCR-CD3 MHC class I immune synapse as an ideal model to compare base editing approaches due to the presence of multiple spliced genes that comprise these complexes (Fig. 2a), the necessity of every gene for surface expression, (Fig. 2b)[28], the ease of a functional readout for biallelic disruption at the single cell level through flow cytometry, the ability to screen guides in a native

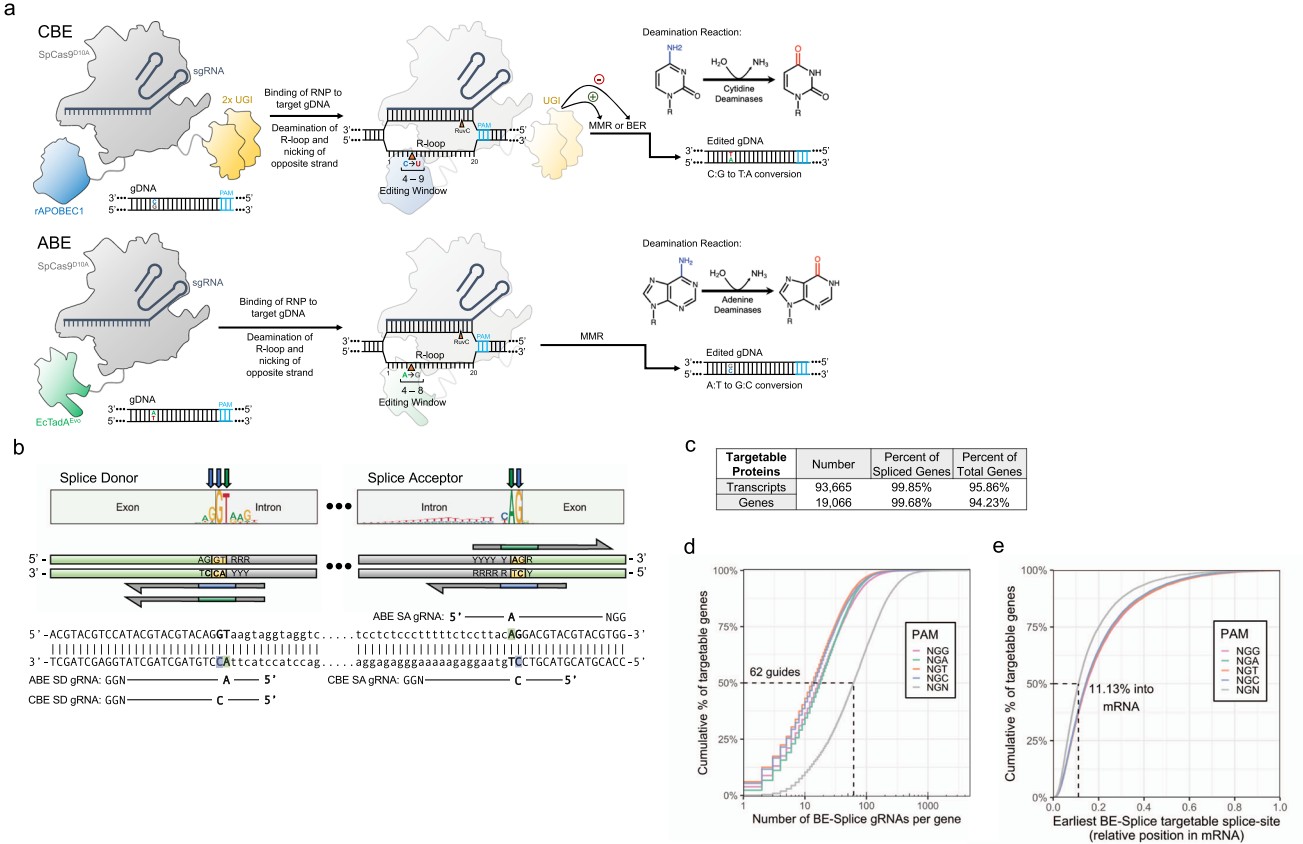

**Fig. 1 Overview of the BE-splice approach. a** Generalized base editing mechanisms of CBEs and ABEs. **b** Positioning of BE-splice sgRNAs within conserved splice-donor and splice-acceptors motif. Logo plots were generated from all human protein coding gene splice sites. Arrows indicate the base targeted by either CBEs (blue), or ABEs (green). **c** Breakdown of transcripts and genes targetable by BE-splice, showing the vast majority of spliced genes are targetable by this approach (99.68%). **d** Distribution of BE-splice sgRNA density across each gene. 50% of genes have 62 or more sgRNAs mapping to them when accounting for all PAM identities and both CBE and ABE approaches. **e** Distribution of the position of the first sgRNA for each gene, with 50% having their first sgRNA 11.13% way through the mRNA or earlier. Source data are available in the Source Data file.

setting, and the interest of the complex in the context of immunotherapies[29–33].

To validate this model, we first designed Cas9 nuclease sgRNAs to each gene in the TCR-CD3 complex and the β2M subunit of the MHC class I complex. Primary human T cells were transfected with TCR-CD3 targeting sgRNAs and Cas9 nuclease, and indel formation was measured via Sanger sequencing, while protein expression was measured via flow cytometry (Supplementary Fig. 4). We found a high level of indel formation (M ± SD, 81.7% ± 15.1%) and high corresponding protein loss (M ± SD, 73.4% ± 37.2%) across all complex members, including CD3ζ (Fig. 2c). These results established that any one of the genes in the TCR-CD3 complex can be edited to induce a loss in surface TCR-CD3 expression.

Next, we wanted to apply our model to directly compare TCR-CD3 disruption mediated by (1) CBEs vs. ABEs, (2) targeting splice donors vs. splice acceptors, and (3) disrupting splicing with BE-splice vs. pmSTOP introduction. We used SpliceR to generate a screen of BE-splice sgRNAs targeting the TCR-CD3 complex, and the iSTOP database[20] to design pmSTOP sgRNAs. Primary human T cells were electroporated with an sgRNA and either the fourth generation CBE, BE4, or the seventh generation ABE, ABE7.10[34]. Samples exhibited a wide range of editing at the target base (M ± SD, 39.6% ± 31.6%, range 0–100%), and a wide range of protein loss (M ± SD, 25.1% ± 31.9%, range 0–97.0%) (Fig. 3a). Both editing of the target base and loss in CD3 surface expression were higher among BE4 than ABE7.10 treated

samples (Student's two-tailed $t$-test, $t = 2.87$, $df = 76$, $P = 5.3e-3$, CI: 6.1–33.7%; $t = 3.75$, $df = 76$, $P = 3.4e-4$, CI: 12.5–41.0%) (Fig. 3b, c).

Within BE4 treated samples, there was no significant difference in editing of the target base among splice donors, splice acceptors, and pmSTOPs (Student's two-tailed $t$-test, SD vs. SA, $t = 0.416$, $df = 38$, $P = 0.680$, CI: $-15.5$%–24.1%), (Student's two-tailed $t$-test, SD vs. pmSTOP, $t = +0.846$, $df = 54$, $P = 4.01e-1$, CI: $-9.9$%–24.4%), (Student's two-tailed $t$-test, SA vs. pmSTOP, $t = +1.27$, $df = 54$, $P = 2.04e-1$, CI: $-6.3$%–29.0%) (Fig. 3b). Furthermore, the rate of protein loss was not significantly different between BE4 splice donors and BE4 splice acceptors (Student's two-tailed $t$-test, $t = +1.27$, $df = 38$, $P = 2.13e-1$, CI: $-8.5$%–36.8%); however, splice donors had significantly higher protein loss efficiencies relative to pmSTOP sgRNAs (Student's two-tailed $t$-test, $t = 3.33$, $df = 54$, $P = 1.57e-3$, CI: 11.3%–45.5%), while splice acceptors were nonsignificantly different than pmSTOP sgRNAs (Student's two-tailed $t$-test, $t = 1.76$, $df = 54$, $P = 8.41e-2$, CI: 2.0%–30.5%). Protein loss was well correlated with disruption of the splice-donor and splice acceptor sites (Two-tailed $t$-test of Pearson's correlation coefficient, $r = 0.843$, $P = 3.11e-06$; $r = 0.65$, $P = 1.92e-3$), yet poorly correlated with the introduction of premature stop codons (Two-tailed $t$-test of Pearson's correlation coefficient, $r = 0.374$, $P = 2.47e-2$) (Fig. 3d). In contrast, ABE7.10 splice-donor sgRNAs had significantly higher editing compared to ABE7.10 splice acceptors (Welch's two-tailed $t$-test, $t = +3.25$, $df = 34.9$, $P = 2.57e-3$, CI: 9.7%–42.0%), which

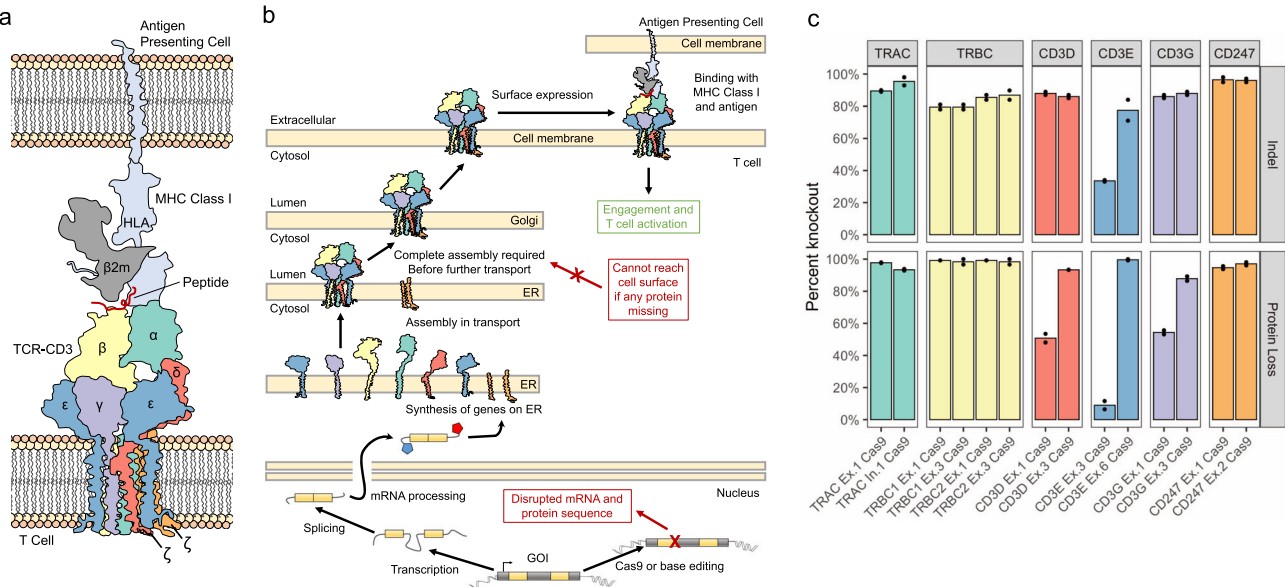

**Fig. 2 Conception and validation of the TCR-CD3-MHC Class I immune synapse as a screening model for protein disruption. a** Diagram of the multimeric TCR-CD3 complex and MHC Class I immune synapse containing multiple spliced genes, based on the solved structures (PDB 6JXR[44], PDB 3T0E[45]; PDB 1OGA[47]). **b** Diagram of the synthesis and localization of the TCR-CD3 complex and interaction with MHC Class I. All members of the CD3 complex are required before functional localization to the cell surface, where disruption of a single splice site within one gene member can prevent a surface expressed complex from forming. **c** Cas9 nuclease knockout of each individual member of TCR-CD3 complex validates the screening model. Two Cas9 nuclease sgRNAs were designed to exonic regions of each gene in the complex. All genes had at least one guide with ≥85% indel efficiency and loss in TCR-CD3 surface expression. Height of bars represents mean of N = 2 independent donors. Source data are available in the Source Data file.

translated into significantly higher protein loss (Welch's two-tailed t-test, $t = +2.81$, $df = 23.9$, P = 9.72e-3, CI: 4.8%–31.6%). Furthermore, protein loss among ABE7.10 treated samples was well correlated with target base editing in splice donor (Two-tailed t-test of Pearson's correlation coefficient, $r = 0.739$, $P = 3.73e{-}05$), but not in splice acceptor sgRNAs (Two-tailed t-test of Pearson's correlation coefficient, $r = 0.506$, $P = 6.51e{-}2$). Among the well correlated BE-splice approaches; BE4 splice donors, BE4 splice acceptors, and ABE7.10 splice donor, the correlation was similar to that observed by Cas9 nuclease (Two-tailed t-test of correlation coefficient, $r = 0.678$, $P = 2e{-}05$) (Fig. 3d). Ultimately, these results established that among the targeted genes (1) CBE mediated protein disruption by BE4 is more reliable than ABE protein disruption mediated by ABE7.10, (2) disrupting splice donors tended to produce more reliable protein disruption across CBEs and ABEs, with a greater disparity observed in ABEs, and (3) the BE-splice method more reliably disrupted the TCR-CD3 complex than pmSTOP introduction.

**Meta-analysis of rAPOBEC1-BE4 and ABE7.10 context dependencies**. With these differences in mind, we wanted to investigate what may cause the disparities in editing efficiency among the different approaches. Previous works established that base editing efficiency is context dependent, with particular preference dictated by the nucleotide preceding the target base[35–37]. These works also sought to change the context dependencies of the preceding nucleotide by employing cytidine deaminase paralogs, orthologs, and engineered variants to change the context dependencies of base editors. Therefore, understanding the dinucleotide context dependencies of base editing would aid in the selection of BE-splice sgRNAs.

To determine these dependencies, we performed an meta-analysis for both BE4 and ABE7.10 with data across multiple cell types, genes, and delivery methods from the literature[19,35,36,38] and data generated by our group (6 papers, 102 guides, 447 edits in total). We chose to focus our analysis on editing efficiency as a

function of the position in the protospacer and the identity of the preceding base. Meta-analysis of BE4 across all nucleotide contexts produced a smooth distribution of editing activity centered about position 6 of the protospacer (Fig. 4a). Consistent with previous work, the editing window was dependent on the identity of the preceding nucleotide[18,25,37,39], where TC dinucleotides exhibited the broadest editing window, while AC and CC exhibited smaller, comparable windows and GC exhibited a highly suppressed window with ≥ 20% activity only observed at positions 5–6 (Fig. 4a). When comparing TC to GC dinucleotides, the identity of the preceding nucleotide alone decreased the average editing efficiency by 3.2-fold from 28.7 to 6.9% (Supplementary Fig. 5a). In contrast, Meta-analysis of ABE7.10 yielded a narrower window with ≥20% editing activity between positions 4 and 8, along with thinner tails to the editing distribution (Fig. 4b). Interestingly, ABE7.10 exhibited similar preceding nucleotide context dependencies as BE4, with TA having the broadest and tallest window, followed by CA, AA, and then GA (Fig. 4b). When comparing TA to GA dinucleotides, the identity of the preceding nucleotide alone decreased the average editing efficiency by 2.9-fold from 24.5 to 6.3% (Supplementary Fig. 5b). Among both BE4 and ABE7.10, the postdinucleotide base did not appear to have as large of an effect on editing efficiency (Supplementary Fig. 5c–f).

Given the distinct differences in editing among dinucleotide contexts, we hypothesized that this could partially explain the differences in baseline rates of editing of the BE-splice sgRNAs. To make this comparison we generated a consensus pentanucleotide motif for each approach we tested for base editing-mediated protein disruption (Fig. 4c). Consistent with the expectation among CBE splice-site motifs, we observed an ACN motif for splice donors, NCT motif for splice acceptors, and a NCA motif for pmSTOPs. The lack of enhancing TC motifs and inhibitory GC motifs across these approaches likely accounts for non-significantly different baseline rates of editing observed across all CBE treated samples. In contrast, ABE splice donors exhibited a

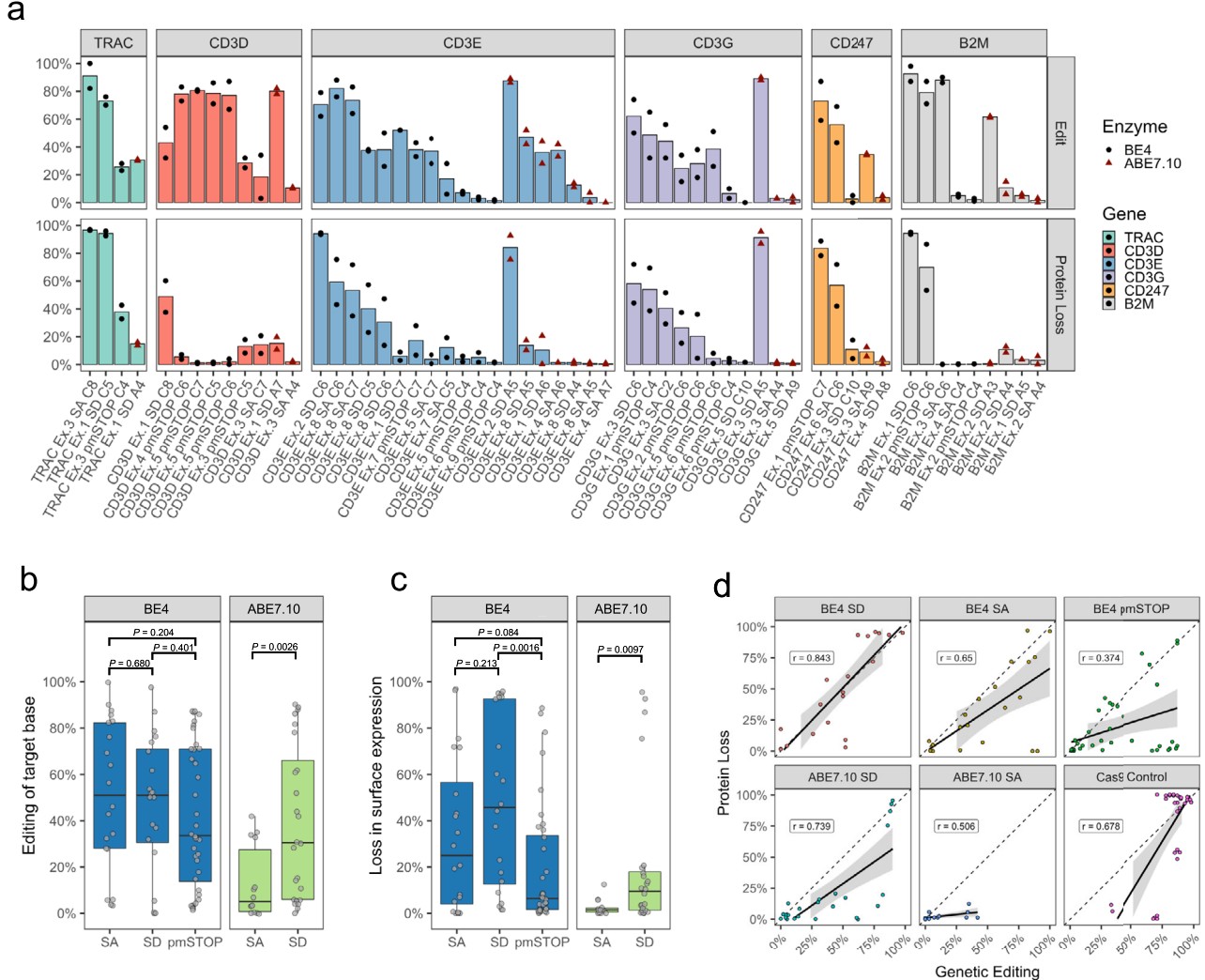

**Fig. 3 BE-splice sgRNAs mediate robust editing and disruption of TCR-CD3 MHC Class I immune synapse. a** Editing efficiency (top) and surface protein loss (bottom) from each guide in the sgRNA screen. Results grouped by gene and enzyme used in descending order by protein loss. *X*-axis label indicates position of target base within sgRNA. TRBC1 and TRBC2 were omitted from the BE-splice screen due to the inability to design single BE-splice sgRNAs to target both paralogs simultaneously. All edits represent the efficiency of target editing; C:G-to-T:A for CBE, and A:T-to-G:C for ABE. Height of bars are mean of replicates. **b, c** Base editing efficiencies or protein loss efficiencies grouped by enzyme and target motif. Data analyzed with Student's two-tailed *t*-test if variance was equal, or Welch's two-tailed *t*-test if variance was unequal with exact *P*-values shown. Boxplot center lines represent the median, box limits represent the upper and lower quartiles, and whiskers define the 1.5× interquartile range. **d** Consistency of editing efficiency and protein loss across all approaches employed here. Relationship between protein loss and base editing efficiency is comparable to that observed in Cas9 control. Error bands represent 95% CI of the mean. All data is from *N* = 2 independent donors, performed on different days. Source data are available in the Source Data file.

preferred T<u>A</u>C motif, while the acceptors exhibited a nondisfavored C<u>A</u>G motif, which likely contributed to the significant difference in baseline rates of editing among ABE treated samples (Fig. 3c).

Underpinning the importance of understanding the nucleotide context preferences of base editors, during the final preparations of this manuscript, Arbab & Shen et al. performed a comprehensive base editor target library analysis to elucidate the determinants of base editing outcomes[40]. With this data they trained a machine learning model (BE-Hive) for the accurate prediction of base editing efficiencies. Comparisons of the observed editing efficiencies in the meta-analysis to predicted editing efficiencies by BE-Hive yielded identical trends in dinucleotide context preference (Supplementary Fig. 6a, b), while supporting our empirical findings that BE4 is on average more

efficient that ABE7.10 (Wilcoxon rank sum test, W = 5.4e4, P = 9.20e-3) (Fig. 4d). Using learnings from BE-Hive and our meta-analysis, we developed a simplified scoring algorithm (Honeycomb) to score BE-splice sgRNAs based on the surrounding sequence context and the position of the target base in the protospacer (Supplementary Fig. 6c–f). We found that Honeycomb and BE-Hive ranked base edits comparably and were well correlated with each other (Fig. 4e), indicating that the simplified Honeycomb algorithm is sufficient for ranking sgRNAs.

**Positional effects of base editor mediated protein disruption.** Next, to investigate how the position of the sgRNA within the transcript affects the reliability of a disruption, we binned all base editing sgRNAs into first, second, middle, second-to-last, and last

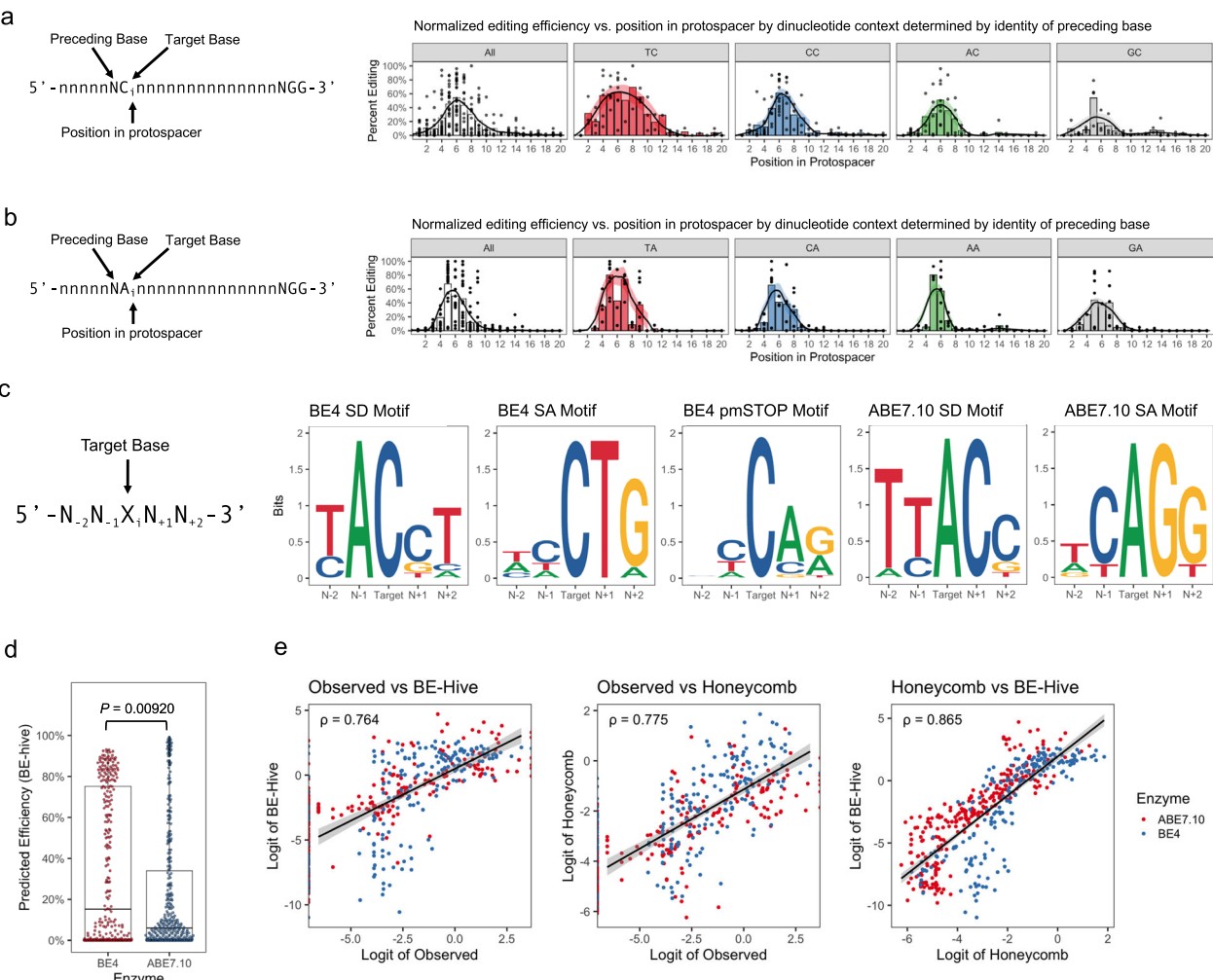

**Fig. 4 Context dependencies of base editors and BE-splice target motifs. a** Dinucleotide context dependencies of rAPOBEC1-BE4. Dinucleotide context is defined by the identity of the target base, and the identity of the base immediately preceding the target. Results normalized and aggregated across published results and our own work. Smoothed distributions generated using LOESS regression with span = 0.5. **b** Preceding dinucleotide context dependencies of TadA$^{WT}$-TadA$^{Evo}$-ABE7.10. Results normalized and aggregated across published results and our own work. Smoothed distributions generated using LOESS regression with span = 0.45. $N = 6$ papers, 102 guides, and 447 edits in the analysis. **c** Logo plots of the pentanucleotide motif for each enzyme and target motif combination in this work. The pentanucleotide motif is oriented with respect to the sgRNA protospacer, independent of how a protospacer is oriented with respect to the direction of gene transcription. The heights of bases are proportional to the prevalence of the base at that position in the target site. **d** Boxplot of BE-Hive predicted editing efficiencies between BE4 and ABE7.10 sgRNAs. Boxplot center lines represent the median, box limits represent the upper and lower quartiles, and whiskers define the 1.5× interquartile range. Analyzed by two-sided Wilcoxon rank sum test due to the non-normal distribution of data. $N = 275$ BE4 edits, and 342 ABE7.10 edits. **e** Comparisons between observed values in meta-analysis and BE-Hive or Honeycomb predicted scorings. Data plotted on a logit scale to better observe relationship of data. Spearman's rank correlation coefficient ($\rho$) is shown. Trend line is linear model line of best fit, grey shading is 95% CI of the mean. Source data are available in the Source Data file.

exons based on which exons they targeted (Supplementary Fig. 7). Strikingly, we found the highest degree of correlation between protein loss and base editing among guides that targeted middle exons (Two-tailed $t$-test of correlation coefficient, $r = 0.915$, $P = 1.6e-20$), and no significant correlation among guides that targeted the last exon (Two-tailed $t$-test of correlation coefficient, $r = 0.359$, $P = 3.09e-1$), as might be expected (Fig. 5a). Next, we wanted to see how the error in protein loss, defined as the absolute value of the observed protein loss minus the editing of target base, varied as a function of the exons being targeted. We found that the error was minimized when targeting inner exons, and increased in guides targeting the second-to-last and last exons (Fig. 5b). Given the extracellular and transmembrane domains of the TCR-CD3 complex are essential to interchain interactions that assemble the complex[41], allow for surface

localization[28], and signal transduction[42–44], we wanted to study how the positioning of these sgRNAs affected the reliability of protein disruption. To do this, we mapped each sgRNA within the tertiary structure of the TCR-CD3 complex (PDB 6JXR[41]) and MHC Class I complex (PDB 3T0E[45]; PDB 1OGA[46]), as determined by the solved structures (Supplementary Figs. 7–14). From these mappings, we classified each sgRNA as extracellular, intracellular, or transmembrane targeting. Consistent with the functional role of these regions, we found that guides targeting the transmembrane region of genes in the complex had the highest correlation of protein loss to editing (Two-tailed $t$-test of Pearson's correlation coefficient, $r = 0.966$, $P = 1.7e-3$, albeit $n = 6$), followed by extracellular guides (Two-tailed $t$-test of Pearson's correlation coefficient, $r = 0.734$, $P = 1.18e-11$), and intracellular guides (Two-tailed $t$-test of Pearson's correlation coefficient, $r =$

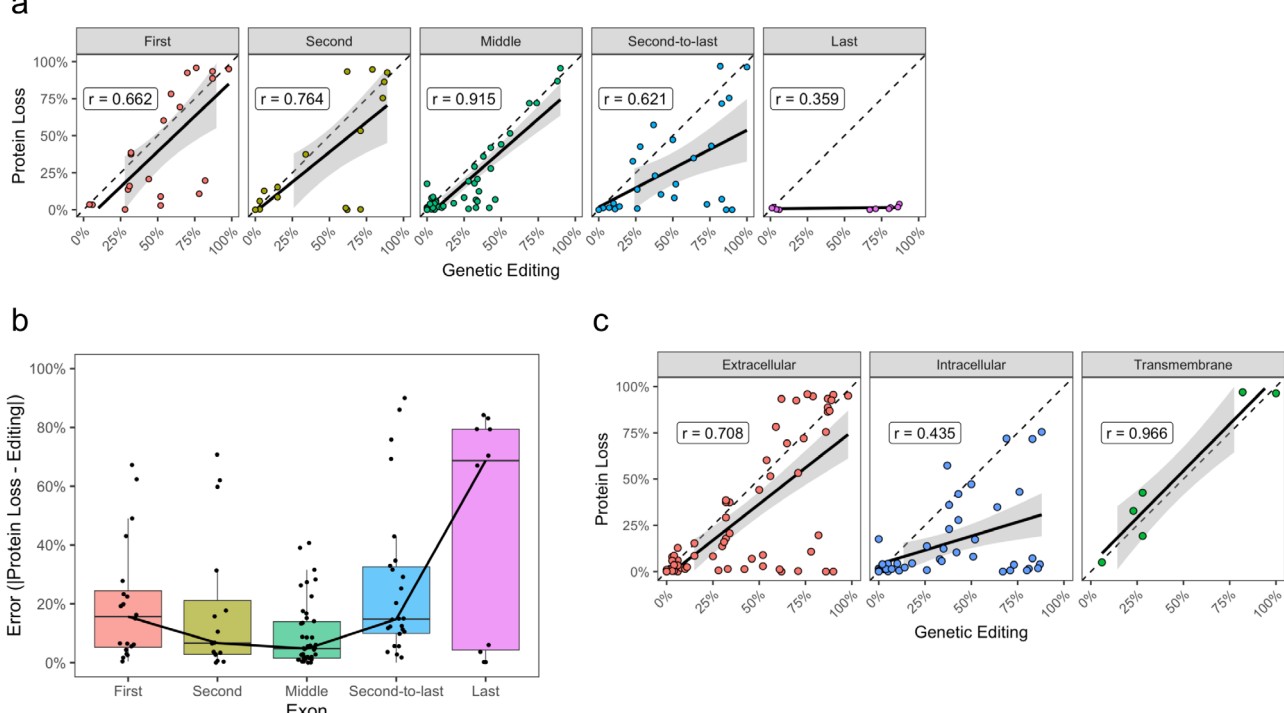

**Fig. 5 Consistency of editing efficiency and protein loss across mRNA and protein regions. a** Scatter plots of protein loss and editing efficiency by exon grouping across all base editor approaches employed. The strongest relationship is observed among middle exons, while the weakest is observed in the last exon. Pearson's correlation coefficient (*r*) is shown. Error bands represent 95% CI of the mean. **b** Error in protein loss as a function of editing efficiency across each exon group. Boxplot center lines represent the median, box limits represent the upper and lower quartiles, and whiskers define the 1.5× interquartile range. The least error is observed among middle exons, while the greatest is observed in the last exon. *N* = 57 unique enzyme-guide combinations with two independent donors. **c** Scatter plots of the protein loss and editing efficiency grouped by where each BE sgRNA maps to the TCR-CD3 and MHC Class I structures. sgRNAs that map to transmembrane and extracellular regions exhibit the greatest consistency between protein loss and base editing efficiency. Pearson's correlation coefficient (*r*) is shown. Error bands represent 95% CI of the mean. Source data are available in the Source Data file.

0.435, *P* = 3.14e-3) (Fig. 5c). Ultimately, these results suggest that maximal reliability of base editing-mediated disruption is achieved when targeting early or inner exons within regions known to be functionally crucial.

**ABE8e allows for enhanced ABE mediated protein disruption.** Last, two recent reports from Richter et al. and Gaudelli et al. describe the development of enhanced eighth generation ABEs, termed ABE8s. Respectively, these works employed either phage assisted directed evolution[47] or large scale mutation libraries[48] to produce ABEs with substantially higher on-target editing. To determine if ABE8s could reduce the disparities we observed between ABE7.10 and BE4 mediated disruption, and to confirm that BE-splice can work for intracellular protein encoding genes, we designed a focused panel of sgRNAs targeting the intracellular immunohibitory gene *CISH*[49] (Fig. 6a, b). We found that ABE8e[47] produced drastic increases in editing efficiency at the DNA level relative to ABE7.10, putting it on par with BE4 treated samples (Fig. 6c, Supplementary Fig. 15). Analysis of *CISH* mRNA by RT-qPCR (Fig. 6d) and whole-cDNA amplification (Fig. 6e) confirmed that higher genetic editing produced proportional disruption in splicing, corroborating mRNA disruption data from the TCR-CD3 screen (Supplementary Fig. 16). Further, a digital western blot for CISH on an unselected population of cells demonstrated a nearly complete loss of protein expression across multiple sgRNAs (Fig. 6f, g, Supplementary Fig. 17). Overall, these results demonstrate that enhanced ABEs can greatly expand the capabilities of BE-splice by increasing base

editing efficiency, and that BE-splice approach can extend to intracellular gene targeting by disrupting splicing patterns.

## Discussion

In this work, we studied the use of CRISPR-Cas9 base editors for highly efficient gene disruption in primary and immortalized human cells using CBEs and ABEs. In this approach, the conserved splice donor (exon|GT-intron) and splice acceptor (intron-AG|exon) sites are edited via a transition mutation (C:G-to-T:A, or A:T-to-G:C), inactivating the splice-site and disrupting the gene at the transcriptional level[22,24]. To improve accessibility to the BE-splice approach, we developed the program SpliceR (z. umn.edu/spliceR) as an online tool for the design of BE-splice sgRNAs. Analysis of the entire human genome showed that 95.86% of all protein coding genes, and 99.85% of all protein coding genes that undergo splicing are targetable. We assessed these predictions, and compared the BE-splice approach to pmSTOP sgRNAs[20] with a mid-throughput screen targeting the TCR-CD3 MHC Class I immune synapse, and with a focused panel targeting the immunoinhibitory intracellular protein CISH. From these experiments, we found three main trends.

First, fourth generation CBEs mediated more reliable disruption than seventh generation ABEs. However, newer eighth generation ABEs appear to reduce this disparity. Consistently, the higher rates of protein disruption may be primarily attributed to the higher levels of DNA editing. The higher editing efficiency of BE4 was consistent with the larger activity window of BE4 relative ABE7.10 observed in our meta-analysis of these enzymes. Furthermore, across all dinucleotide contexts, BE4 exhibited a

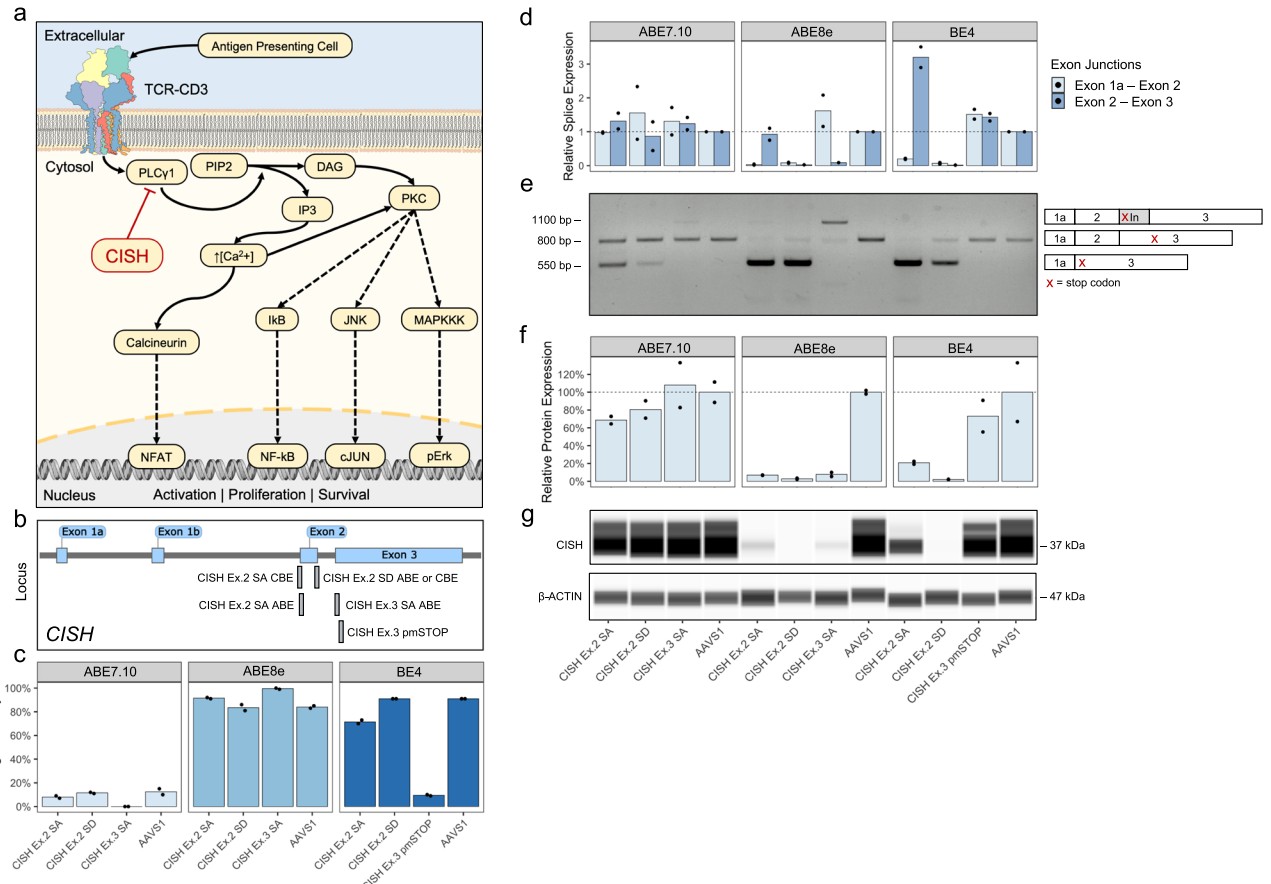

**Fig. 6 Using ABEs and CBEs to disrupt the intracellular protein CISH in the K562 cell line. a** Diagram of CISH immunohibitory pathway. **b** Mapping of sgRNAs to CISH locus. **c** Editing efficiencies with sgRNAs paired with ABE7.10, ABE8e, and BE4. AAVS1 is an inert locus control. **d** Taqman expression assays of CISH exon boundaries. Data normalized to AAVS1 control. $N = 2$ biological replicates, each with 3 technical replicates. **e** Representative gel image of whole-cDNA amplification of edited samples with altered isoforms from two independent biological replicates. See Supplementary Fig. 15 for uncropped gel image. **f** Relative protein expression quantified from digital western blot of CISH. CISH expression normalized within-samples to β-ACTIN and between-samples to AAVS1 control. Height of bars represents mean of $N = 2$ biological replicates. **g** Representative digital western blot of CISH and β-ACTIN from two independent biological replicates. See Supplementary Fig. 17 for uncropped western blot images. Source data are available in the Source Data file.

smoother, more normally distributed window compared to ABE7.10, which may be the result of the processive activity of the CBE deaminase, *R. norvegicus* APOBEC1. In contrast, the ABE7.10 deaminase was evolved from *E. coli* TadA, which acts endogenously on a single adenosine within the tRNA$^{Arg}$ anticodon loop[50,51].

Second, across CBEs and ABEs, splice donor targeting produced the most reliable protein disruption. In CBEs, splice donors and splice acceptors were not significantly different in editing efficiency, while in ABEs, splice donors had a significantly higher editing efficiency. These trends were also mirrored in the rate of protein disruption with both enzymes, with the rate of protein loss being nonsignificantly different in CBE splice donors relative to splice acceptors, and significantly higher among ABE splice donors relative to splice acceptors. Among ABE7.10, this disparity could be attributed in part to the highly preferred TAC motif in splice-donor guides, and the lesser preferred CAG motif in splice acceptor guides. The TAC preference motif is consistent with the preference motif of the parental enzyme *E. coli* TadA, and other adenosine deaminases[50–52].

Additionally, previous work has demonstrated that targeting the splice-donor sequence with an indel forming Cas9 nuclease causes robust protein and lncRNA knockout[53,54]. The reliability of disrupting the splice donor may be explained by the critical

nature of the splice-donor site in initiating splicing. During splicing, pre-mRNA nucleotides in the splice donor define the exon boundary and initiate splicing through Watson-Crick-Franklin base pairing rules between the pre-mRNA and the U1 or U12 snRNP[55,56]. Therefore, if this RNA:RNA duplex is disrupted at the outset of splicing, then the ability to undergo the native splicing event may be principally inhibited.

Moreover, when considering the enhanced reliability of targeting splice donors from the splice acceptor perspective, previous work demonstrates that disrupting splice acceptors tends to favor exon skipping, which may retain the reading frame as opposed to introducing a nonsense outcome[22]. Further, work studying the biological plasticity of Cas9 nuclease introduced frameshift mutations shows that exon skipping in edited cells can result in protein isoforms with internal sequence deletions that retain some biological function[57]. Conversely, as demonstrated by the multiple instances of clinical splice-site mutations[58,59], it is important to note that splice acceptor mutations do not always result in a clean, single exon skipping outcome, as previous publications have suggested[22]. Rather, splice acceptor mutations are capable of inducing alternative splicing patterns via activation of cryptic splice sites, such as full intron retention, partial intron retention, and partial skipping of an exon[60–62], all of which have the potential to disrupt the function of a gene by introducing a

frameshift mutation or by removing a functionally critical region of the molecule.

Third, among the genes in our screen, CBE splice sgRNAs produced more frequent protein disruption than pmSTOPs sgRNAs. At the genetic level, there was no significant difference in editing efficiency of BE-splice guides compared to pmSTOP guides. However, at the protein level, pmSTOPs produced significantly less disruption compared to splice donors, and non-significantly different disruption compared to splice acceptors. Consistent with our analysis of CD3E RNA from treated samples (Supplementary Fig. 16), these disparate results may be attributed to different levels of induction of nonsense mediated decay. Premature stop codons are known to incompletely induce a nonsense mediated decay[63], or they can form relatively functional truncated variants[64]. In contrast, there is evidence that one-third of human genes have alternative isoforms that are actively suppressed by nonsense mediated decay[65]. This suggests that if a base-edited splice site induces an alternative isoform that naturally occurs due to basal splicing errors, then these isoforms may be more readily subjected to nonsense mediated decay than newly introduced premature stop codons which have not been encountered by the cell.

Conversely, recent work by Hanna et al.[59] modeling loss-of-function variants using massively paralleled base editor screens, suggests that when analyzing splice-site targeting sgRNAs in aggregate (i.e., splice donors and splice acceptors grouped together), there is a similar efficacy in the proportion of splice-site targeting sgRNAs vs. pmSTOP introducing sgRNAs that generate loss-of-function variants. However, their data also show that splice-site targeting sgRNAs tend to generate more extreme loss-of-function Z-scores than pmSTOP introduction sgRNAs. This suggests that splice-site targeting guides may have a greater expressivity of disruption, which may explain some of the disparities we observe between BE-splice and pmSTOP approaches.

Regardless of the mechanisms at play, the average efficiencies of both BE-splice and pmSTOP approaches could both be improved by increasing the baseline rates of editing with more active CBEs that have decreased or altered sequence preferences, such as EvoFERNY-BE4[37] or hA3A-BE4[39]. Moreover, it is important to note that in our view and application of using base editing for gene disruption we see BE-splice and pmSTOP introduction as complementary approaches.

A standardized workflow for designing and testing BE-splice sgRNAs aids in the expedience, and reliability of validating sgRNAs (Supplementary Fig. 18). When designing BE-splice sgRNAs with SpliceR it is important to use the Ensembl transcript table to choose whichever transcript or transcripts are most relevant to the biological phenomenon of interest. In instances where disruption of the established functions of a gene are desired, we recommend targeting transcripts with merged Ensembl automated and Havana manual annotation ("gold labelled"). Furthermore, our results demonstrate that targeting all exons besides the last exon can produce high-efficiency disruption, where the innermost exons produce the most proportional protein loss to genetic editing efficiency. We recommend initially screening 3–4 guides per gene with a greater than 50% predicted editing efficiency. If additional CBE sgRNAs are desired, we also recommend including high predicted efficiency, inner exon pmSTOP sgRNAs to a known functional region from the iSTOP database. Whole-cDNA amplification of the target transcripts aids in validating the effect of the sgRNA at the mRNA level to observe if a change occurs in the isoform pool.

Furthermore, given situations where none of the aforementioned approaches are available, we also recommend trying to model known pathogenic variants such as those documented on ClinVar, Uniprot, or COSMIC for generating loss-of-function

variants. However, as is the case with Cas9 nuclease editing, with any base editing approach (Fig. 7)[15,20–22,24,59,66,67] we recommend functionally validating the effect of a base edit at the protein level, rather than assuming high editing efficiency DNA editing corresponds to a loss-of-function.

Lastly, much excitement has been generated by the prospect of prime editing[68], which in principle allows for the precise editing and induction of mutations 80 bp or smaller in size, including transversion, transition, and indel mutations. The ability to target a wide variety of mutations lends prime editing potential superiority over base editing in the correction of pathogenic mutations for gene therapy. This advantage is mainly derived from the ability of prime editing to precisely introduce desired mutations without undesired bystander edits to adjacent bases in the editing window as is seen with base editing. However, in the context of disrupting a gene by editing a conserved element, unintended bystander edits are of less concern when the goal is to inactivate a splice site. Furthermore, despite the low-to-moderate level of indels observed in prime editing, this method still needs to be evaluated in a multiplex setting where compounding indels will likely lead to translocations[68]. The effect of these indels will also need to be addressed in stem cells, where double strand breaks have been associated with reduced potency[68,69] and an increased risk of oncogenesis through inhibition of regulators the cell cycle and genomic stability[12,70].

As more is understood about the consequences of Cas9 nuclease induced DSBs, it is of increasing interest to deploy gene editing methods that do not rely on DSBs. Here we show that CBEs and ABEs can allow for high-efficiency disruption of proteins in primary cells by disrupting conserved splice-sites. Collectively, our results inform the application of base editing for protein disruption, and in selecting optimal BE-splice guides generated by SpliceR. Ultimately, we believe that the BE-splice approach is a widely applicable technique and holds particular promise in the sensitive landscape of cell based therapies.

## Methods

**T-cell isolation.** Peripheral blood mononuclear cells (PBMCs) from healthy, adult donors were purchased from Memorial Blood Center (St. Paul, Minnesota). PBMCs were isolated with the Trima Accel leukoreduction system (LRS, Memorial Blood Center) chambers using ammonium chloride-based red blood cell lysis. Upon receipt of PBMCs, cells were further purified for CD3 + cells by immuno-magnetic negative selection using the EasySep Human T-cell Isolation Kit (STEMCELL Technologies, Cambridge, MA). T cells were frozen at $10–20 \times 10^6$ cells per 1 mL of Cryostor CS10 (STEMCELL Technologies) and thawed into culture for editing experiments.

**Cell culture.** Primary human T cells were cultured in OpTmizer CTS T-cell Expansion SFM (ThermoFisher, Waltham, MA) containing 2.5% CTS Immune Cell SR (ThermoFisher), L-Glutamine, Penicillin/Streptomycin, N-Acetyl-L-cysteine (10 mM, Sigma–Aldrich, St. Louis, MO), IL-2 (300 IU/mL, PeproTech, Rocky Hill, NJ), IL-7 (5 ng/mL, PeproTech), and IL-15 (5 ng/mL, PeproTech) at 37 °C and 5% $CO_2$. Prior to electroporation T cells were activated with Dynabeads Human T-Activator CD3/CD28 (ThermoFisher) at a 2:1 bead:cell ratio for 48 h. Following electroporation, T cells were maintained at $1 \times 10^6$ cells/mL in a 24-well or 12-well plate. K562 cells (ATCC CCL-243) were cultured at a density of $5 \times 10^5$ cells per mL in RPMI with 10% FBS and 1× Penicillin/Streptomycin. Cells were kept at 37 °C and 5% $CO_2$.

**sgRNA design.** For each gene of interest, the major isoform to design sgRNAs against was identified using convergent Ensembl, Havana, Uniprot, RefSeq, and GENCODE annotations. For Cas9 nuclease sgRNAs, the isoform transcript ID was queried through the Synthego Knockout Guide Design tool (https://design.synthego.com/#/) and guides were chosen based on having a high predicted on-target efficiency[71] and low predicted off-targets. For TRAC, Cas9 sgRNAs previously validated sgRNAs were used[72]. BE-splice sgRNAs were designed using SpliceR (z.umn.edu/splicer) and pmSTOP sgRNAs were designed using the iSTOP database (http://www.ciccialab-database.com/istop). Both BE-splice and pmSTOP sgRNAs were selected based on having ≥20% maximal editing efficiency, using the dinucleotide context and the position of the target base within the protospacer, as

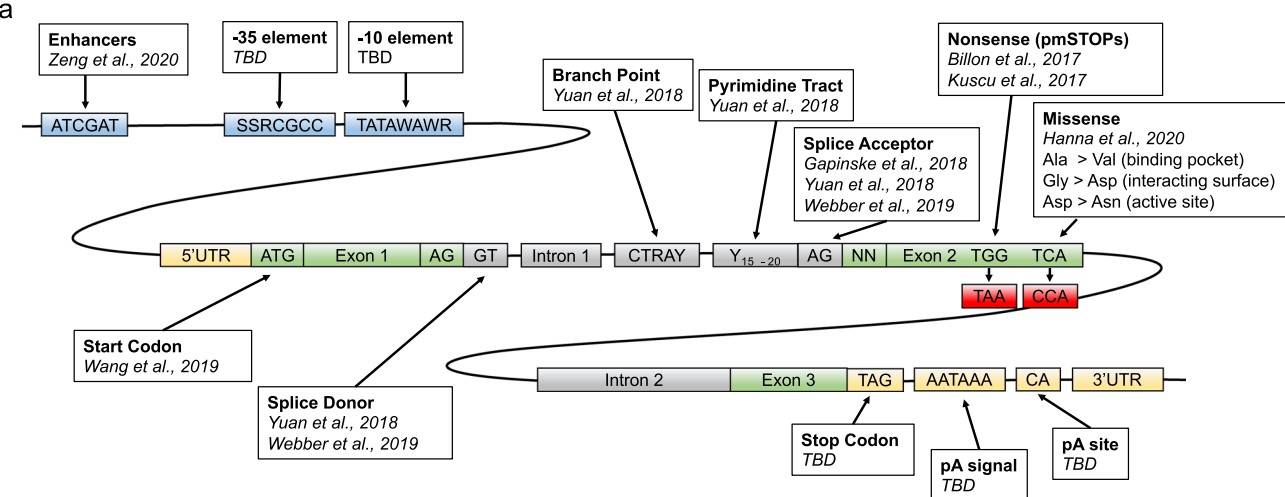

**Fig. 7 Disrupting genes with base editors. a** Conserved genetic elements that can be targeted for gene disruption by base editors. Elements that have been targeted in publications are cited, while additional conserved elements that have not been validated at the time of this publication are indicated as to be determined (TBD). Elements include enhancers[66], −35 element, −10 element, start codon[67], Splice donor[15,24], Branch point[24], Pyrimidine tract[24], Splice acceptor[15,22,24], Nonsense[20,21], Missense[59], Stop codon, pA signal, pA site. Source data are available in the Source Data file.

detailed in Komor et al.[18] and Gaudelli et al.[19]. All sgRNAs and corresponding genomic DNA primers used in this study are available in Supplementary Fig. 19.

**Primary human T-cell and K562 transfection**. Forty-eight hours following T cell activation, Dynabeads were magnetically removed, cells were washed once with PBS, spun down, and resuspended in P3 Primary Cell Nucleofector™ Solution containing Supplement 1 (Lonza, Basel, Switzerland). Ten million T cells were combined with 1 μg of chemically modified sgRNA (Synthego, Menlo Park, CA) and 1.5 μg codon optimized BE4 mRNA (TriLink Biotechnologies, San Diego, CA), codon optimized ABE7.10 mRNA (TriLink Biotechnologies), or codon optimized *Sp*Cas9 mRNA (TriLink Biotechnologies). T cells were electroporated with the 4D-Nucleofector system (Lonza) using a P3 16-well Nucleocuvette kit, with $1 \times 10^6$ T cells per 20 μL cuvette using the program EO-115. T cells were allowed to recover in Nucleocuvettes for 15 minutes before being transferred to 300 μL of antibiotic-free CTS OpTmizer T cell Expansion SFM medium at 37 °C, 5% $CO_2$. Twenty minutes after transfer, 700 μL of complete CTS OpTmizer T cell Expansion SFM was added to the culture. K562 (ATCC® CCL-243™) were similarly transfected. Briefly, cells were cultured at a density of $5 \times 10^5$ cells per mL. On the day of electroporation, $2 \times 10^5$ K562 cells were resuspended in SE buffer with Supplement 1 and combined with 1 μg of chemically modified sgRNA, and 1.5 μg of enzyme mRNA. Cells were electroporated with the 4D-Nucleofector system using a SE 16-well Nucleocuvette kit, with $2 \times 10^5$ cells per 20 μL cuvette using the program FF-120. Fifteen minutes following electroporation, cells were transferred from the nucleocuvette to 1 mL of K562 media.

**Flow cytometry**. Seven days after electroporation $5 \times 10^5$ T cells were collected and stained in 100 μL with fluorophore-conjugated anti-human antibodies for CD3 (BD Biosciences #564001), beta-2-microglobulin (BioLegend #316306) and Fixable Viability Dye eFluor780 or LIVE/DEAD Fixable Aqua Dead Cell Stain (Thermo-Fisher #65-0865-14 1:500 dilution, #L34966 1:40 dilution) were used to assess cell viability. Events were acquired on LSR II or LSRFortessa flow cytometers using FACSDiva software, and data was analyzed using FlowJo v10 software. See Supplementary Fig. 4 for gating strategy. The percent of positive events were normalized, by dividing over the sample percent positivity over pulse alone control percent positivity. Protein loss was then calculated as [1—% normalized sample positivity].

**Genetic analysis**. Seven days after electroporation genomic DNA was isolated from T cells or K562 by spin column-based purification (GeneJET, ThermoFisher). Editing efficiency was analyzed on the genomic level by PCR amplification of the targeted loci, Sanger sequencing of the PCR amplicons (ACGT or Eurofins Genomics). Sanger sequencing traces of base-edited samples were analyzed using EditR (z.umn.edu/editr)[73], while indel efficiency was analyzed using the Synthego ICE program (https://ice.synthego.com/#/)[74]. See Supplementary Fig. 19 for guide protospacer and primer sequences.

**Taqman expression assays**. RNA was isolated from cell pellets using RNeasy Mini Kit (Qiagen, Hilden, Germany). RNA concentration was measured with Nanodrop One (Thermo Fisher Scientific) and stored at −80 °C. First-strand cDNA was synthesized with SuperScript III First Strand Synthesis SuperMix for

qRT-PCR kit (Thermo Fisher Scientific). 500 ng of RNA was reverse transcribed with both oligo(dT)$_{20}$ and random hexamers. RT-PCR was performed on a Mastercycler (Eppendorf, Hamburg, Germany) using cycling conditions at 25 °C for 10 min, 50 °C for 30 min and 85 °C for 5 min. After cycling, RNAse H treatment was used to remove the RNA template. cDNA was then diluted 1:5 prior to use in qRT-PCR. The qRT-PCR was conducted using Applied Biosystems TaqMan Universal PCR Master Mix and Taqman Assays following the manufacturer's procedure (Thermo Fisher Scientific). In brief, 30 ng of first-strand cDNA was amplified in triplicate, with both human housekeeping gene GAPDH Taqman assay and target human CISH Taqman assays. Thermocycling program was 40 cycles of 95 °C for 15 s and 60 °C for 1 min with an initial cycle of 50 °C for 2 min and 95 °C for 10 min. All amplifications and detections were carried out in a MicroAmp optical 96-well reaction plate with CFX96 Real time System (Bio-Rad Laboratories, Hercules, CA). The Comparative Ct ($2^{-\Delta\Delta Ct}$) method was used to analyze the relative gene expression level, which was normalized to the housekeeping gene and relative expression measured relative to the control sample using equation (1).

$$2^{\Delta\Delta C_t} = 2^{\left(\frac{\left(C_{t,\,sample,\,GOI} - C_{t,\,sample,\,\beta-actin}\right)}{\left(C_{t,\,control,\,GOI} - C_{t,\,control,\,\beta-actin}\right)}\right)} \quad (1)$$

**Isoform analysis**. cDNA was synthesized from RNA using the aforementioned procedure. Five μL of 1:5 diluted cDNA was used as a template for the PCR reaction. The PCR was performed according to the manufacturer's protocol using Accuprime Taq polymerase (Thermo Fisher Scientific) and primers designed to amplify all exons of the target isoforms with the cycle; [94 °C—2:00], 30 × [94 °C—0:30, 55 °C—0:30, 68 °C — 0:30], [68 °C—5:00], [4 °C—hold]. PCR products were run on a 2% agarose-TAE gel with ethidium bromide and imaged using a Bio-Rad Universal Hood II Gel Documentation System. Gel bands were excised and DNA was extracted using QIAquick Gel Extraction Kit (Qiagen), followed by Sanger sequencing with the PCR primers (ACGT or Eurofins Genomics).

**Immunoblot analysis**. Protein was isolated from $1 \times 10^6$ T cells or K562 cells using RIPA buffer containing protease inhibitors (Sigma–Aldrich, COEDTAF-RO, P5726, and P0044) for 30 min at 4 °C. A 4 °C spin at $12,000 \times g$ for 15 min was performed and the supernatant was extracted for BCA quantification (Thermo Fisher Scientific). The JESS was used according to the manufacturer's protocol for capillary electrophoresis and protein immunoblotting. From the digital electropherogram, relative protein expression was calculated using equation (2).

$$Relative\ Protein\ Expression = \frac{\left(\frac{AUC_{sample,\,CISH}}{AUC_{sample,\,\beta-actin}}\right)}{\left(\frac{AUC_{control,\,CISH}}{AUC_{control,\,\beta-actin}}\right)} \times 100\% \quad (2)$$

**SpliceR development and whole-genome guide prediction**. SpliceR was written in the statistical programming language R (v. 3.6.1) using RStudio (v. 1.1.383). The SpliceR web app was developed using R shiny (https://shiny.rstudio.com/). All files are available online (https://github.com/MoriarityLab/SpliceR); dependencies.R is responsible for installing the necessary libraries to run the application; global.R and

helpers.R define the functions needed to interact with Ensembl, and generate BE-splice sgRNAs, while server.R and ui.R communicate with each other to handle the user inputs, generate sgRNAs, and return the output to the user interface. SpliceR relies on tidyverse (https://www.tidyverse.org/) and Bioconductor (https://www.bioconductor.org/) packages. For the whole-genome guide prediction, the *runSpliceR()* function from the SpliceR web app (v.2.0.0) was modified to run as a command line executable. All human protein coding gene Ensembl transcript IDs were pulled from the GENCODE database (https://www.gencodegenes.org/human/). The *runSpliceR()* function was then run in parallel across all Ensembl transcript IDs using the Minnesota Supercomputing Institute.

**BE4 and ABE7.10 context dependency analysis**. Papers employing the CBE rAPOBEC1-BE4, and the ABE TadA^WT-TadA^Evo-ABE7.10 were found using PubMed and Google scholar. Using a combination of main and supplementary figures the editing values for each targetable base (adenosines for ABEs, and cytidines for CBEs) within the protospacer was manually recorded. Data generated from our own work was similarly entered. All manual data entries were double entered over two independent times to check for consistency of values. To control for variable electroporation and baseline rates of editing efficiencies across different works, each editing value was normalized to the maximum editing efficiency observed in each cell type, for each work. To account for the same guide being used in different studies, normalized percent editing was averaged across each position of each unique guide. Data were then analyzed based on the predinucleotide, and postdinucleotide context of each base. R script for reproducible analysis is available in Source Data file.

Prediction of base editing efficiency was performed using the BE-Hive model[40] with the 'be_predict_pystander' package (https://github.com/maxwshen/be_predict_bystander). Each protospacer with flanking −20 and +10 genomic bases were analyzed by the 'bystander_model' function with 'celltype' parameter set to "HEK293T" and 'base_editor' parameter set to "ABE" or "BE4". Results were processed with source code in the Source Data file.

**Data analysis and visualization**. All statistical analyses were performed in R using RStudio. The level of significance was set at $\alpha = 0.05$. All statistical tests were first subjected to assumptions of homoscedasticity. For samples with equal variance, Student's two-sample, unpaired two-tailed *t*-test was used, while for samples with unequal variance Welch's two-sample, unpaired two-tailed *t*-test was used. Wilcoxon rank sum test was used for samples with non-normal distributions. Pearson's correlation coefficient ($r$) was used, except for when daily was highly skewed in which case Spearman's rank correlation coefficient ($\rho$) was used (Fig. 4e). Data were visualized in R studio employing various tidyverse (https://www.tidyverse.org/), Bioconductor (https://www.bioconductor.org/) packages, and ggseqlogo (v0.1) (https://github.com/omarwagih/ggseqlogo) packages[75]. See Source Data file for reproducible analyses and full software version details.

**Reporting summary**. Further information on research design is available in the Nature Research Reporting Summary linked to this article.

## Data availability

GENCODE protein coding gene annotations available online (ftp://tp.ebi.ac.uk/pub/databases/gencode/Gencode_human/release_37/gencode.v37.annotation.gff3.gz). PDB structures (https://www.rcsb.org/) PDB 6JXR43, PDB 3T0E47, and PDB 10GA48 were used in this work. Raw sequencing reads are available via NCBI BioProject accession number PRJNA702523. All other data are available from the authors upon reasonable request. Source data are provided with this paper.

## Code availablity

Code and data for figure reproduction are found in the Source Data file. SpliceR web app source code is available through GitHub (https://github.com/MoriarityLab/SpliceR).

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

## Acknowledgements

We thank Dr. Matthew Johnson at the University of Minnesota for discussions surrounding flow cytometry. We thank Dr. Kevin Holden at Synthego for insight about chemically modified sgRNAs. This work was funded by the Children's Cancer Research Fund, NIH grant R03AI144840, and the University of Minnesota Academic Investment Research Program.

## Author contributions

M.G.K., W.S.L., B.S.M., and B.R.W. designed experiments. M.G.K., W.S.L., C.L.L., B.A.S., X.Q., N.J.S., P.N.C., E.J.P., M.J.V., S.C.L., S.P.B., and A.A.A. performed experiments. M.G.K., W.S.L., C.L.L., S.P.B., S.P.P., and S.C.L. analyzed the data. M.G.K. wrote the program. M.G.K., W.S.L., B.S.M., and B.R.W. wrote the manuscript. All authors approved and read the final manuscript. These authors contributed equally: M.G.K. and W.S.L. These authors contributed equally: B.R.W. and B.S.M.

## Competing interests

The authors declare the following competing interests: B.R.W. and B.S.M. are consultants for Beam Therapeutics. B.R.W and B.S.M. have financial interests in Beam Therapeutics. M.G.K., W.S.L., C.L.L., E.J.P., B.R.W., and B.S.M. are inventors of a full patent *Lymphohematopoietic engineering using cas9 base editors* (WO2019178225A2), which covers the application of using base editing for gene disruption in lympho-hematopoietic cells. All Author's interests were reviewed and are managed by the University of Minnesota in accordance with their conflict of interest policies. The Authors B.A.S., X.Q., N.J.S., P.N.C., S.P.B., S.P.P., S.C.L., and A.A.A. declare no competing interests.
