## [Peer Review File · Nature Communications]

Reviewers' Comments:

Reviewer #1:

Remarks to the Author:

Kluesner and Lahr et al. describe a new tool called SpliceR and provide a direct comparison of several gene knockout techniques in primary human T-cells. The SpliceR program interfaces with the ensemble database to generate and score sgRNAs for BE-mediated knockout (KO) by splice site mutation. The authors target the TCR-CD3 immune synapse using wt-Cas9 nuclease, CBEs, and ABEs as a promising method to reduce alloreactivity. They show gene KO of any subunit via Cas9-mediated indel introduction is sufficient to functionally disrupt surface expression of the TCR-CD3 complex. Additionally, this manuscript shows KO by CBE is more reliable than ABE and KO via splice donors>splice acceptors>pmSTOP. Additionally, they show that targeting the middle exon produces the most reliable protein loss results. Finally, they include a discussion with considerations for BE-mediated KO approaches.

Overall, I believe the experiments are thoughtful and rigorous, and the results will be of interest to the field. I do have some comments/concerns though.

Major:

1. Although this is the first comprehensive analysis comparing different ABE, CBE, and Cas9 nuclease knockouts, the novelty is over-stated throughout the paper. The abstract states there has been no direct comparison between premature stops and splice sites, but the Webber and Lonetree et al. 2019 paper does exactly that (compares pmSTOP/SA/SD). Similarly, statements #2 and #3 at end of intro were already known for CBE.
2. For Fig. 5/pg. 7, the discrepancy between gene editing and protein KO may be explained by biological plasticity (Smits et al. 2019 doi.org/10.1038/s41592-019-0614-5) such as exon skipping or translation reinitiation. Rationalizing the efficient protein KO for splice donor disruption would benefit the manuscript. Would the non-sense mediated decay pathway degrade these transcripts without the opportunity for alternative splicing/plasticity?
3. Why did the authors mainly target exon 1 and 2 sites for their wtCas9 experiments? It has been well-established that for genes with various isoforms, middle exons should be targeted for knock-out studies (which the authors validate with their data in Figure 3a). If no major isoforms for the TCR-CD2-MHC complex exist, this should be stated. Either way, additional gRNAs should be included for the wtCas9 work to allow for a more valid comparison with the BE work.
4. I would be hesitant to state that the results of this study can be broadly applied to anyone who wishes to perform a gene knock-out study in the future. While these results are incredibly relevant to immunotherapy work, given that all genes that were targeted in this work are part of the same complex, and only one cell type was used, I don't know how transferable these results are to other genes and/or cell types.
5. The Gaudelli et al. ABE8 paper tests more gRNAs (24 vs. 19 in the current work) for protein knock-out via ABE-mediated splice site disruption in primary T-cells (with both ABE7.10 and a variety of ABE8s), which the authors do not mention in their manuscript. Additionally, Gaudelli et al demonstrate that ABE8 can edit splice sites with 98-99% efficiency, which is not mentioned in the current manuscript, and again reduces the novelty of the current work.

Minor:

1. On pg. 2 in the first paragraph, describing base editors as installing C:G to T:A or A:T to G:C (as done in the discussion) will display the ability to target the opposite strand for complimentary base conversions. In addition to replication, DNA repair pathways may preserve the edited base / convert the base pair into the intended product.
2. The acronym NSESI is confusing and does not seem to be established in literature. Are the genes excluded from analysis all single-exon genes or intronless genes (Jorquera et al. 2018 doi.org/10.1093/database/bay089)? Finding a different way to sufficiently describe this would clarify what pool of genes are 99.85% targetable (all spliced genes?).
3. Fig 1b coloring the protospacers and denoting the PAM with something other than an arrow may help the reader.

4. Does figure 3 editing efficiency only account for C to T and A to G edits? Quantifying the undesired CBE edits (C to non-T) at the pmSTOP locations edited in these experiments, or the previous iSTOP/CRISPR-STOP papers, would add to the significance of BE-splice methods.
5. There have been analysis of ABE and BE4 sequence preference individually with similar results as shown in this manuscript. However, the comprehensive/meta-analysis and pentanucleotide consensus motif do add to the body of knowledge of base editor sequence preference.
6. Adding that ABE8s and next-generation CBEs have decreased or altered sequence preferences in the discussion would be useful.
7. Ref. 38/51 are now published in Nature Comm/Nature Biotech, respectively.
8. Figure 1a is confusing with the multiple C's shown in the R-loop. None of the target sites the authors use have five C's in a row, and this would be specific to only CBEs, while the figure is meant to generically refer to both CBE and ABE

Reviewer #2:

Remarks to the Author:

This paper reports two main findings

- 1) A new programme to search for base editing sites to target splice sites (donor and acceptor).
- 2) In then evaluates some of predictions by using ABE and CBE in context of disrupting TCR/APC interactions by targeting both splice sites and stop codons.

It is generally well written, but needs a number of amendments and corrections as listed below under headings of general and specific.

In addition, a clarification is required regarding the definition of the "preceding base" (P6-L15) which is required before I can accurately complete the review of the second half of this manuscript.

GENERAL COMMENTS

The title of the paper refers to disruption of proteins, but throughout document mainly uses protein knock out. In figure 3, authors refer to loss of surface expression. I think title is fine, but in main text, they should consider replacing "protein knock out" with "protein disruption" and/or "loss of surface expression throughout document".

Also, they refer to gene knockout frequently in the document, when base editing makes very precise single base changes at precise gene structures, so wonder if terms such as "targeted splice site inactivation" or similar would be better description of work. Knock out usually refers to a gross disruption; base editing doesn't do that (and in on figure, they use "null" which means whole gene is removed – see specific comment below).

A small amount of additional information is required before detailed evaluation of some aspects of data can be reviewed in detail. This comment relates to use of term "preceding base" in second half of manuscript (P6-L15). This term isn't defined in the paper, it doesn't occur in any of the four references cited (P6-L18). It's not defined in Figure 1 either (which would be a good place to highlight it). Thus, until this is clarified, I can't comment in detail about latter half of results (which look very interesting, but I need to be clear which base is being referenced).

Another small thing to make manuscript more accessible is to modify figure 1 to fully clarify how ABE and CBE are being used to target these splice sites and stop codons. I am very familiar with the field, but Figure 1B shows arrows pointing to G and T nucleotides on top strand of DNA, when in fact they should be pointing to C and A (respectively) on bottom strand. A second clarification is

required for logo diagrams in figure 4 as in three cases out of four, they are made for strand on which deamination event occurs, rather than strand consensus signal for splicing is usually referred to. They are not wrong, but a minor change would make them much easier for readers to follow.

One other comment regarding overall manuscript that affects protein disruption is that there appears to be no mention of whether targeting occurs on one or both alleles for any of the genes selected? I assume all target genes are autosomal, so both copies would need to be inactivated for substantial loss of protein (and essential if authors insist on use of knockout). I would request a comment on this in the manuscript.

My final general comment is that authors mistakenly use terms such as "the rate of loss was non-significantly higher" (P9-L20.21) on several occasions – a rate of loss that was non-significantly higher is not a loss, and should be referred to correctly as "no-significant difference was observed".

SPECIFIC COMMENTS

ABSTRACT

P1-L26 – they need to mention cell(s) and proteins on other side of the synapse, e.g. antigen-presenting cells/MHC

P1-L29 – maybe rephrase slightly to say BE is better at targeting splice donors than splice acceptors

INTRODUCTION

P2-L9/10 – insert "to edit the other DNA strand" after "template" – as written, doesn't quite convey role of nickase and DNA repair process?

P2-L15 – rephrase such that statement becomes and/or – at present could be misread that a stop codon and a splice change are needed to inactivate.

P2-L34 – the use of protein knockout is incorrect, editing targets gene (I've suggested it's inactivation not knockout) and the protein then simply isn't made properly. A minor rephrasing is suggested.

P2-L38 – see earlier general comment about phrase gene knockout

RESULTS

P3 – L12 – the phrase "these results suggested ... IS a robust method" is a slight over-statement. The authors use results to PREDICT that BE will be a robust method, so please rephrase the second part of the statement as a prediction.

P3-L13 – based on above, "appearing" should be changed to "potentially"

P3-Figure 1 – this gives nice overview of the BE-splice approach but 1a with the Cas9, gRNA and DNA target is upside down relative to other DNA in the figure, and doesn't include PAM site. The presence of the Cas9 protein distracts from the more useful info on location of PAM site relative to editing window (neither of which are labelled in the diagram), so minor simplification of figure

would help (that part could actually be omitted) as line diagram below table in 1A has the requisite info. See comments above about request to change arrows such that actual base being deaminated is correctly shown.

P3-L28 – see earlier about protein knockout

P4-Figure 2a – this is a very nice diagram capturing the immune synapse with great clarity (my only minor criticism is regarding the zeta chain labelling which could be missed. Could it be repositioned very slightly such that it's not mistaken for a squiggle (at first I thought it was last few amino acids of the the cartoon, then I realised it was the name)).

P4-Figure 2b – coding sequence is not "null" (which implies total gene deletion), rather it's coding sequence with PTC. In middle of diagram, it should read cannot reach surface if any PROTEIN missing. Text in lower part of figure is very small.

P5-L1 – "not significantly higher than" should be replaced with "no different from" or "the same as"

P5-Figure 3 mentions loss of protein expression (at cell surface); this may be a better term than protein knockout.

P6-L2 – replace "baseline editing" with another term to avoid potential confusion with "base editing"

P6-15 – see general comments above, but "preceding base" needs to be unambiguously defined. Is it before the target residue – maybe, but what if several target residues in same editing window.

NOTE – It is not possible for me to complete review of second part of paper until clarified. The clarification is required as data looks potentially very exciting, and I would like to be able to review it accurately.

P7-Figure 4c – see general comments regarding the logo plots. At first glance, one would expect to see agGTa for a splice acceptor sequence, not tACct. However, I realise it is referring to lower strand in opposite orientation, but this may not be obvious to general reader. It is then confusing as the BE4 SA and ABE SA are presented in same manner, but the ABE SD is in the orientation one would expect. Redrawing these figures will make it much easier to follow.

P9-L31 – I don't want to undermine author's efforts to accurately reflect Rosalind Franklin's contribution to the scientific understanding DNA structure, but am not aware that "Watson-Crick-Franklin interaction" is a recognised term. "Watson-Crick interaction" is the correct term, a reference appearing in the mid 1970's based on a pubmed search in reference to the 1953 Nature paper by Watson and Crick.

Reviewer #3:

Remarks to the Author:

CRISRP/Cas composite gene knockout technologies based on adenosine or cytosine deaminases (collectively termed "base editors") are highly popular approaches both academically and commercially, and have a number of advantages to direct CRISRP/Cas9 approaches. Here, the authors compare stop codon mediated, gene knockout technologies based (DNA) base editing, with splice site disruption based genic disruption.

The core of the paper is formed by an algorithm that picks specific sgRNAs to base-edit splice

sites. This algorithm is then used in a head to head comparison of CBE vs ABE editors and point mutation vs splice site deletion efficiencies in knockout generation. The system in which this comparison takes place is the multiplex editing of the T cell immune synapse (a very clever approach since loss of TCR/MHC can be determined by FACS whereas loss of the gene in its various configurations is determined by Sanger sequencing).

Running the wet lab experiment was extremely informative (probably beyond what the authors initially intended). Interestingly they found that targeting splice donor sites produced substantially more robust knockouts than pmSTOPs (in itself perhaps not unexpected); that targeting middle exon splice sites was far more efficient than either 5' or 3' exons; and that targeting splice donor sites was superior to targeting acceptor sites. Perhaps due to the sequence surrounding SD and SA sites, and the unique preferences of each of the base editors, CBEs (rAPOBEC1-BE4) were more reliable than ABEs (ABE7) in generating splice site based protein knockouts.

Overall this is an excellent paper with robust validation of a proposed tool - but beyond that, with the wet lab providing interesting novel insights that actually lead to optimization of the proposed approach. And beyond: one could argue (e.g fig 5) that they provide inadvertently an excellent window into the mechanism of splicing and into how genic disruption leads to protein loss.

Response to Reviewers

Dear Reviewers,

Thank you for your review of our manuscript, #NCOMMS-20-14319A, entitled, “**CRISPR-Cas9 cytidine and adenosine base editing of splice-sites mediates highly-efficient disruption of proteins in primary and immortalized cells**”, for publication in Nature Communications. We are grateful for the insightful comments and critiques and have addressed them below in a point-by-point manner. Revisions to the manuscript text are colored in red text.

The following new data has been added to the revised manuscript:

- Comparison of our meta-analysis on the editing context dependencies of BE4 and ABE7.10 to values predicted by the recently developed machine learning model BE-Hive (Fig. 4e-d). Further, we expand upon our learnings from the meta-analysis and learnings from Arbab & Shen et al., to develop a simplified algorithm for scoring and ranking BE-splice sgRNAs by SpliceR, which we call Honeycomb (Fig. 4e, Supplementary Fig. S6). When examining sgRNAs in our meta-analysis we find that this algorithm allows for reliable scoring, and it is well correlated with predictions made by BE-Hive while having the advantage of requiring considerably less memory, and is computationally faster.
- Comparison of ABE7.10, ABE8e, and BE4 in targeting the intracellular immunohibitory gene *CISH* in both K562 and T cells (Fig. 6, Supplementary Fig. S17). Data demonstrates that 1) ABE8e reduces the disparity between ABEs and CBEs to mediate disruption, 2) intracellular genes can be disrupted with high efficiency by BE-splice at the DNA, RNA and protein level, and 3) confirms that BE-splice works in multiple cell types.
- Quantitative RT-PCR of CD3E and CISH showing that protein loss due to splice site targeting can be explained in some samples by NMD induction (Supplementary Fig. 16), or exon skipping that result in a truncated reading frame (Fig. 6).

REVIEWER COMMENTS

Reviewer #1 (Remarks to the Author):

Kluesner and Lahr et al. describe a new tool called SpliceR and provide a direct comparison of several gene knockout techniques in primary human T-cells. The SpliceR program interfaces with the ensemble database to generate and score sgRNAs for BE-mediated knockout (KO) by splice site mutation. The authors target the TCR-CD3 immune synapse using wt-Cas9 nuclease, CBEs, and ABEs as a promising method to reduce alloreactivity. They show gene KO of any subunit via Cas9-mediated indel introduction is sufficient to functionally disrupt surface expression of the TCR-CD3 complex. Additionally, this manuscript shows KO by CBE is more reliable than ABE and KO via splice donors>splice acceptors>pmSTOP. Additionally, they show that targeting the middle exon produces the most reliable protein loss results. Finally, they include a discussion with considerations for BE-mediated KO approaches.

Overall, I believe the experiments are thoughtful and rigorous, and the results will be of interest to the field. I do have some comments/concerns though.

Major:

1. Although this is the first comprehensive analysis comparing different ABE, CBE, and Cas9 nuclease knockouts, the novelty is over-stated throughout the paper. The abstract states there has been no direct comparison between premature stops and splice sites, but the Webber and Lonetree et al. 2019 paper does exactly that (compares pmSTOP/SA/SD). Similarly, statements #2 and #3 at end of intro were already known for CBE.

We thank the reviewer for their considerations on framing the novelty of our work. We revised the manuscript to refocus the novel aspects of our work, and included additional experiments and analyses that reflect recent developments in the field.

First, in regards to claim #2 and #3 at the end of the introduction (p.2, l.40-44), to our knowledge this is the first controlled investigation testing the hypothesis that splice site targeting with base editors more reliably achieves disruption of protein function. While the data in Webber et al suggests those trends, the experiments in that work were not designed to address this question. In contrast, this work was specifically conceived to test this hypothesis through properly powered, balanced, and controlled experiments. Furthermore, by targeting the recently solved TCR-CD3, we were able to map guides at the protein and genetic level, lending insight to the importance of guide positioning in disruption, and the necessity of certain domains in disruption.

Second, we reframed the manuscript to focus on these additional novel aspects:

- The development of modular, easy-to-use tool that is adaptable to a continually expanding suite of base editor variants being used in many organisms
- A meta-analysis of the base editing literature we find strong, convergent evidence of the importance of the base editing context in dictating editing outcomes

- An independent comparison of our meta-analysis to the recently developed BE-Hive algorithm which yielded good agreement. This allowed for the further development of a BE-splice scoring algorithm (Honeycomb) which we implemented in SpliceR (p.8, l.40-44; p.9, l.1; p.10, l.1-9) (Fig. 4, Supplementary Fig. S6).
- Use of eight generation ABE8e to show that it can eliminate disparities observed between CBE and ABE approaches.
- Showing that splice-site targeting can disrupt intracellular genes that are therapeutically relevant, such as CISH which is being targeted in ongoing CRISPR-Cas9 clinical trials ([NCT04426669](https://clinicaltrials.gov/ct2/show/study/NCT04426669)) (p.11-12) (Fig. 6, Supplementary Fig. S15, S17).

2. For Fig. 5/pg. 7, the discrepancy between gene editing and protein KO may be explained by biological plasticity (Smits et al. 2019 doi.org/10.1038/s41592-019-0614-5) such as exon skipping or translation reinitiation. Rationalizing the efficient protein KO for splice donor disruption would benefit the manuscript. Would the nonsense mediated decay pathway degrade these transcripts without the opportunity for alternative splicing/plasticity?

We thank the reviewer for their interpretation of the results and their recommended article. We modified the manuscript to discuss our results within the idea of biological plasticity (p.14, l.1-3). Further, to empirically address the role of NMD decay we provide additional analysis of CD3E targeted samples (Supplementary Fig. S16). Although we found the strongest evidence of NMD in the lead guide, CD3E Ex.2 SD, loss of protein expression was not strictly predicated on evidence NMD (e.g. CD3E Ex.8 SA C6, and CD3E Ex.8 SA C7). Therefore, NMD may frequently explain protein loss by eliminating an opportunity for biological plasticity to occur, however it does not appear strictly necessary to induce a loss of function. We included an expanded discussion of this topic in the discussion (p.13, l.42 - p.14, l.10).

3. Why did the authors mainly target exon 1 and 2 sites for their wtCas9 experiments? It has been well-established that for genes with various isoforms, middle exons should be targeted for knock-out studies (which the authors validate with their data in Figure 3a). If no major isoforms for the TCR-CD2-MHC complex exist, this should be stated. Either way, additional gRNAs should be included for the wtCas9 work to allow for a more valid comparison with the BE work.

We thank the reviewer for raising questions about the gRNA selection process. Cas9 guide design began by identifying the major isoform for each gene using convergent annotations from Ensembl, Havana, Uniprot, RefSeq, and GENCODE. Based on these annotations, all genes in this TCR-CD3 complex were documented to have a major isoform. The transcript ID of the major isoform was then queried through the Synthego Knockout Guide Design tool and guides were chosen based on having a high predicted on-target efficiency and a low number of predicted off-targets. All genes besides TRAC were designed to have at least one guide targeting an inner exon, which we define as not targeting the first or the last exon. For TRAC this was not the case because we used previously validated sgRNAs ([PMID: 28225754](https://pubmed.ncbi.nlm.nih.gov/28225754/)). To make this process clearer to the reader, a more detailed explanation of guide selection is included in the revised manuscript (p.16, l.41 - p.17, l.8).

Ultimately, the purpose of this experiment was to validate previous studies (PMID: 29261409) showing that every member of the TCR-CD3 gene is required for surface localization. Consistently, we found that high efficiency disruption of each gene independently resulted in the loss of TCR-CD3 surface expression. This confirmed that multiple genes could be independently targeted for a common functional read out, thereby validating the utility of the model for screening purposes.

4. I would be hesitant to state that the results of this study can be broadly applied to anyone who wishes to perform a gene knock-out study in the future. While these results are incredibly relevant to immunotherapy work, given that all genes that were targeted in this work are part of the same complex, and only one cell type was used, I don't know how transferable these results are to other genes and/or cell types.

We thank the reviewer for this consideration. To further investigate the generalizability of splice-site targeting we revised the manuscript to include targeting of an intracellular protein (CISH) in an additional cell type (T cells and K562 cell line) (Fig. 6, Supplementary Fig. S15 and S17, p.11 - p.12). It is also important to note that there is a wide body of literature demonstrating that clinical splice-site mutations induce loss-of-function mutations in a variety of genes, over a variety of tissues (PMID: 29680930), which supports the generalizability of targeting splice sites for gene disruption.

5. The Gaudelli et al. ABE8 paper tests more gRNAs (24 vs. 19 in the current work) for protein knock-out via ABE-mediated splice site disruption in primary T-cells (with both ABE7.10 and a variety of ABE8s), which the authors do not mention in their manuscript. Additionally, Gaudelli et al demonstrate that ABE8 can edit splice sites with 98-99% efficiency, which is not mentioned in the current manuscript, and again reduces the novelty of the current work.

We thank the reviewer for raising questions about the sample size and encouraging a comparison to ABE8s.

First, Supplementary Table S1 of Gaudelli et al. indicates 18 SpCas9, and 6 SaCas9 guide sequences. These guides were used to investigate the activity of each evolved variant from each round of directed evolution — these guides were not used to study splice site disruption. Meanwhile, Supplementary Table S2 of Gaudelli et al. indicates that 6 guides were used to target splice sites for gene disruption in T cells, while 1 guide was used to disrupt the promoters of HBG1 and HBG2 in CD34+ HSCs. In contrast, our work used 64 unique base editor-guide combinations, including 22 unique splice-site targeting ABE guides (additional_files_7). Therefore, in terms of comparing guides for ABE-mediated splice site disruption in T cells, our work has a greater number of guides.

Second, we agree with the importance of using eighth generation ABEs. As such, we included an additional experiment in the revised manuscript comparing ABE8e to ABE7.10, and BE4 demonstrating that ABE8e drastically improves editing efficiency and protein loss. Additional sections of the manuscript were revised to reflect this state-of-the-art (p.12, l.1-7) (Fig. 6).

Minor:

1. On pg. 2 in the first paragraph, describing base editors as installing C:G to T:A or A:T to G:C (as done in the discussion) will display the ability to target the opposite strand for complimentary base conversions. In addition to replication, DNA repair pathways may preserve the edited base / convert the base pair into the intended product.

We revised the manuscript to use the suggested notation (p.2, l.12-13) and to reflect the central role of DNA repair in preserving the edited base (p.2, l.14-16).

2. The acronym NSESI is confusing and does not seem to be established in literature. Are the genes excluded from analysis all single-exon genes or intronless genes (Jorquera et al. 2018 doi.org/10.1093/database/bay089)? Finding a different way to sufficiently describe this would clarify what pool of genes are 99.85% targetable (all spliced genes?).

We appreciate the reviewer seeking clarity on this point, and their recommendation on terminology. In the original manuscript, SESI referred to what Jorquera et al. termed *intronless genes*, therefore the 99.85% of targetable genes are all non-SESI genes, which are not *intronless genes*. To more clearly communicate this we changed all usage of NSESI to “spliced genes” or “genes that undergo splicing” per your suggestion.

3. Fig 1b coloring the protospacers and denoting the PAM with something other than an arrow may help the reader.

We agree with the reviewer and modified Fig. 1b to clearly denote the location of the PAM.

4. Does figure 3 editing efficiency only account for C to T and A to G edits? Quantifying the undesired CBE edits (C to non-T) at the pmSTOP locations edited in these experiments, or the previous iSTOP/CRISPR-STOP papers, would add to the significance of BE-splice methods.

We thank the reviewer for raising this question. Fig. 3 editing efficiencies only account for target editing, i.e. C:G-to-T:A for CBE, and A:T-to-G:C for ABE. The Fig. 3 legend has been revised with this clarification.

Furthermore, we agree with the reviewer that undesired, non-target edits produced by CBEs would still allow splice-site disruption, while they would not allow for pmSTOP introduction and are thus a potential advantage of BE-splice. Historically, this was especially true with earlier generations of CBEs (e.g. non-codon optimized BE3 or BE4), as these enzymes regularly produced these non-target outcomes. However, using codon optimized vectors and an all mRNA delivery system tends to eliminate these problems. For example, by NGS we typically see $\geq 95\%$ product purity of the desired C:G-to-T:A edits using our protocol with chemically modified sgRNAs and codon optimized mRNA for enzyme expression (Webber et al., 2019). As such, for both BE-

splice and pmSTOP, levels of undesired non-targets edits are quite low and are no longer as distinct of an advantage to BE-splice as they were prior to adopting improved protein expression methods.

5. There have been analysis of ABE and BE4 sequence preference individually with similar results as shown in this manuscript. However, the comprehensive/meta-analysis and pentanucleotide consensus motif do add to the body of knowledge of base editor sequence preference.

We thank the reviewer for their comments.

6. Adding that ABE8s and next-generation CBEs have decreased or altered sequence preferences in the discussion would be useful.

We revised the discussion to include this suggestion (p.14, l.34-37).

7. Ref. 38/51 are now published in Nature Comm/Nature Biotech, respectively.

We updated the references to reflect the publication of these manuscripts (p.11, l.1-4).

8. Figure 1a is confusing with the multiple C's shown in the R-loop. None of the target sites the authors use have five C's in a row, and this would be specific to only CBEs, while the figure is meant to generically refer to both CBE and ABE

We agree, and revised Fig. 1a to more accurately depict the substrates of CBEs and ABEs.

Reviewer #2 (Remarks to the Author):

This paper reports two main findings

1) A new programme to search for base editing sites to target splice sites (donor and acceptor).

2) It then evaluates some of predictions by using ABE and CBE in context of disrupting TCR/APC interactions by targeting both splice sites and stop codons.

It is generally well written, but needs a number of amendments and corrections as listed below under headings of general and specific.

In addition, a clarification is required regarding the definition of the “preceding base” (P6-L15) which is required before I can accurately complete the review of the second half of this manuscript.

Thank you for the opportunity to provide further clarification. This change has been incorporated into this version of the manuscript and we look forward to further review.

GENERAL COMMENTS

1. The title of the paper refers to disruption of proteins, but throughout document mainly uses protein knock out. In figure 3, authors refer to loss of surface expression. I think title is fine, but in main text, they should consider replacing “protein knock out” with “protein disruption” and/or “loss of surface expression throughout document”.

We thank the reviewer for stimulating a discussion about a term that is central to conceptualizing our work. We have considered this comment and while we feel that knock-out can refer to the removal of protein function, the usage of the term is ambiguous. Per the reviewers recommendation, we updated the manuscript to use the term “protein disruption” to more accurately reflect the nuances of splice site targeting.

We understand the reviewers point in terms of a classical definition of knock-outs, and the historical methods by which knock-outs, whereby a whole locus or multiple exons were removed from the genome. However, we feel there is ambiguity in the modern use of the term, particularly in recent years as gene knock-outs regularly refer to loss-of-function gene disruption created by introducing small indel mutations with targeted nucleases. Similarly, introducing a splice-site mutation, or premature stop codon, to create an altered or truncated protein product appears to meet this same definition of knock-out used when referring to Cas9 nuclease editing.

However, given we agree with the ambiguity of the term we heed the reviewers comment and replaced knock-out with “protein disruption” or “loss of surface expression” throughout the manuscript with minor exceptions. Those exceptions are in the use of the use of knock-out when referring to Cas9 nuclease induced protein disruption, as it is standard in the field to use the term knock-out in this instance, or the use of the term “functional knock-out” or “functional gene knock-

out” in minor cases (p.1, l.36,39; p.2, l.27; p.13, l.35; p.20, l.23,29) to describe when the function of a protein is eliminated.

2. Also, they refer to gene knockout frequently in the document, when base editing makes very precise single base changes at precise gene structures, so wonder if terms such as “targeted splice site inactivation” or similar would be better description of work. Knock out usually refers to a gross disruption; base editing doesn’t do that (and in on figure, they use “null” which means whole gene is removed – see specific comment below).

We understand the perspective of the reviewer, please refer to our response to point 1. Further we agree that “inactivating splice site” is a precise term and have used it in the revised manuscript (p.1, l.19; p.13, l.6; p.16, l.5).

3. A small amount of additional information is required before detailed evaluation of some aspects of data can be reviewed in detail. This comment relates to use of term “preceding base” in second half of manuscript (P6-L15). This term isn’t defined in the paper, it doesn’t occur in any of the four references cited (P6-L18). It’s not defined in Figure 1 either (which would be a good place to highlight it). Thus, until this is clarified, I can’t comment in detail about latter half of results (which look very interesting, but I need to be clear which base is being referenced).

We thank the reviewer for raising clarification and apologize for any confusion. The “preceding base” refers to the identity of the base (e.g. A,C,G,T) preceding the base being edited, C in the case of CBEs, and the target A in the case of ABEs. To clarify this terminology, we revised Fig. 4 to explicitly indicate what is meant by the preceding base relative to a base of interest. Note that this definition extends any base of interest in the protospacer, and that a single protospacer typically has several bases of interest.

4. Another small thing to make manuscript more accessible is to modify figure 1 to fully clarify how ABE and CBE are being used to target these splice sites and stop codons. I am very familiar with the field, but Figure 1B shows arrows pointing to G and T nucleotides on top strand of DNA, when in fact they should be pointing to C and A (respectively) on bottom strand.

We agree with the reviewer and updated Fig. 1b to explicitly show which base editor, disrupts which splice-site, through which base, on which strand.

5. A second clarification is required for logo diagrams in figure 4 as in three cases out of four, they are made for strand on which deamination event occurs, rather than strand consensus signal for splicing is usually refereed to. They are not wrong, but a minor change would make them much easy for readers to follow.

We thank the reviewer for the suggestion. We modified Fig. 4c to show that all motif diagrams are with respect to the 5'-to-3' context surrounding the target base. The Fig. 4c legend was also

modified to clarify the orientation of the logo diagrams with respect to the protospacer, and with respect to the orientation of transcription.

6. One other comment regarding overall manuscript that affects protein disruption is that there appears to be no mention of whether targeting occurs on one or both alleles for any of the genes selected? I assume all target genes are autosomal, so both copies would need to be inactivated for substantial loss of protein (and essential if authors insist on use of knockout). I would request a comment on this in the manuscript.

We agree with the reviewer that targeting autosomal genes typically requires loss of both alleles to disrupt protein function, underpinning the importance of generating biallelic edits in producing cells with an observable phenotype.

This concern is addressed by our use of flow cytometry, which collects information about protein loss at the single-cell level. If a cell does not exhibit cell surface expression of the marker of interest, such as the TCR-CD3 complex or β 2M, then it does not possess a functional copy of these genes. Therefore, in the case of autosomal genes these cells contain a biallelic edit. Furthermore, given the lead guides for each gene had $\geq 80\%$ genetic editing and $\geq 80\%$ loss of protein, it necessitates that the majority of cells in these populations have biallelic edits. Per the reviewer's request we included a comment on this in the manuscript (p.4, l.8).

6. My final general comment is that authors mistakenly use terms such as "the rate of loss was non-significantly higher" (P9-L20.21) on several occasions – a rate of loss that was non-significantly higher is not a loss, and should be referred to correctly as "no-significant difference was observed".

We agree with the importance of using statistically precise language, as such the manuscript has been revised to reflect this comment (p.13, l.28-30).

SPECIFIC COMMENTS

ABSTRACT

- 1. P1-L26 – they need to mention cell(s) and proteins on other side of the synapse, e.g. antigen-presenting cells/MHC*

All genes studied in this work involved in the TCR-CD3 MHC Class I synapse are expressed on T cells, which both present and recognize antigens. Regardless, this manuscript is not an investigation of immune synapse dynamics, rather it focuses on the disruption of splice sites for protein disruption.

2. *P1-L29 – maybe rephrase slightly to say BE is better at targeting splice donors than splice acceptors*

We agree in not overstating the significance of our findings and adjusted the sentence to acknowledge splice acceptors and introducing premature stop codons. (p.1, l.24-25).

INTRODUCTION

P2-L9/10 – insert “to edit the other DNA strand” after “template” – as written, doesn’t quite convey role of nickase and DNA repair process?

We revised this sentence to clarify the role of the nickase in stimulating DNA repair (p.2, l.14-16).

P2-L15 – rephrase such that statement becomes and/or – at present could be misread that a stop codon and a splice change are needed to inactivate.

Thank you for the comment. We revised the manuscript to reflect this change (p.2., l.22).

P2-L34 – the use of protein knockout is incorrect, editing targets gene (I’ve suggested it’s inactivation not knockout) and the protein then simply isn’t made properly. A minor rephrasing is suggested.

Please see earlier response to general comment 1. We revised the manuscript to reflect this change (p.2, l.39).

P2-L38 – see earlier general comment about phrase gene knockout

Please see earlier response to general comment 1. We revised the manuscript to reflect this change (p.2, l.44).

RESULTS

P3 – L12 – the phrase “these results suggested ... IS a robust method” is a slight over-statement. The authors use results to PREDICT that BE will be a robust method, so please rephrase the second part of the statement as a prediction.

We agree with the reviewer and revised the manuscript accordingly (p.3, l.12).

P3-L13 – based on above, “appearing” should be change to “potentially”

We revised the manuscript accordingly (p.3, l.23-24).

P3-Figure 1 – this gives nice overview of the BE-splice approach but 1a with the Cas9, gRNA and DNA target is upside down relative to other DNA in the figure, and doesn't include PAM site. The presence of the Cas9 protein distracts from the more useful info on location of PAM site relative to editing window (neither of which are labelled in the diagram), so minor simplification of figure would help (that part could actually be omitted) as line diagram below table in 1A has the requisite info. See comments above about request to change arrows such that actual base being deaminated is correctly shown.

Please see earlier response to general comment 4. We revised Fig. 1 accordingly.

P3-L28 – see earlier about protein knockout

We revised this statement accordingly (p.4, l.8).

P4-Figure 2a – this is a very nice diagram capturing the immune synapse with great clarity (my only minor criticism is regarding the zeta chain labelling which could be missed. Could it be repositioned very slightly such that it's not mistaken for a squiggle (at first I thought it was last few amino acids of the the cartoon, then I realised it was the name)).

We bolded the zeta chain in Fig. 2 to make the labelling clearer.

P4-Figure 2b – coding sequence is not “null” (which implies total gene deletion), rather it's coding sequence with PTC. In middle of diagram, it should read cannot reach surface if any PROTEIN missing. Text in lower part of figure is very small.

We thank the reviewer for their comment on Fig. 2. We removed the use of the null, changed to “if any protein missing”, and have eliminated the small text in the figure.

P5-L1 – “not significantly higher than” should be replaced with “no different from” or “the same as”

We appreciate the reviewers comment on use of statistical language and incorporated this change into the manuscript (p.6, l.16-17)

P5-Figure 3 mentions loss of protein expression (at cell surface); this may be a better term that protein knockout.

Please see earlier response to general comment 1. We revised the manuscript to reflect these changes.

P6-L2 – replace “baseline editing” with another term to avoid potential confusion with “base editing”

We agree with the reviewer and replaced the term with “baseline rates of editing”. (p.8, l.29-39)

P6-15 – see general comments above, but “preceding base” needs to be unambiguously defined. Is it before the target residue – maybe, but what if several target residues in same editing window.

Please see earlier response to general comment 3. To expand upon the clarification, this analysis accounts for several editable bases in the same window. For example, if a CBE guide protospacer has the sequence **AC₂TC₄AC₆GC₈TGGATAGTAGGAG**, then there would be 4 potential targets in the window, each with a unique position (e.g. 2, 4, 6, or 8), a unique preceding base (e.g. AC, TC, AC, GC), and an observed editing efficiency. For the meta-analysis, this information was aggregated across all gRNAs to generate the consensus plots in Fig. 4a-b to describe the context dependencies of BE4 and ABE7.10.

NOTE – It is not possible for me to complete review of second part of paper until clarified. The clarification is required as data looks potentially very exciting, and I would like to be able to review it accurately.

P7-Figure 4c – see general comments regarding the logo plots. At first glance, one would expect to see agGTa for a splice acceptor sequence, not tACct. However, I realise it is referring to lower strand in opposite orientation, but this may not be obvious to general reader. It is then confusing as the BE4 SA and ABE SA are presented in same manner, but the ABE SD is in the orientation one would expect. Redrawing these figures will make it much easier to follow.

Please see earlier response to general comment 3. Fig. 4c and the figure legend have been revised to more clearly communicate the orientation of the motifs.

P9-L31 – I don’t want to undermine author’s efforts to accurately reflect Rosalind Franklin’s contribution to the scientific understanding DNA structure, but am not aware that “Watson-Crick-Franklin interaction” is a recognised term. “Watson-Crick interaction” is the correct term, a reference appearing in the mid 1970’s based on a pubmed search in reference to the 1953 Nature paper by Watson and Crick.

We appreciate the reviewer’s attention to using terms that allow for a cohesive and productive scientific discourse. Further, we agree with the reviewer that historically the term “Watson-Crick interaction” has been preferred to refer to the structure of nucleic acid duplexes, however in recent years “Watson-Crick-Franklin interaction” has also become a recognized term. As such, a brief google scholar search points to several examples in reputable journals that use this term to describe the structural interactions of nucleic acid duplexes:

Rauch et al., 2020, J Am Chem Soc
Knutson et al., 2020, J Am Chem Soc
Knutson et al., 2020, J Am Chem Soc
Chien et al., 2020, J Am Chem Soc
Porto et al., 2020, Nat Rev Drug Discov
Hagler et al., 2002, Canadian J Chemistry

Mahdavi-Amiri et al., 2020, Royal Society of Chemistry
Aquino-Jarquin, 2020, Molecular Therapy

Given that 1) there is precedent for using this term, 2) the term communicates the helical structure and interaction between the U1 snRNP and the pre-mRNA splice donor site, and 3) acknowledges Rosalind's Franklin's tantamount contributions to the understanding the structure of nucleic acid duplexes, we have left this term in the revised manuscript (p.13, l.39).

Reviewer #3 (Remarks to the Author):

CRISRP/Cas composite gene knockout technologies based on adenosine or cytosine deaminases (collectively termed "base editors") are highly popular approaches both academically and commercially, and have a number of advantages to direct CRISRP/Cas9 approaches. Here, the authors compare stop codon mediated, gene knockout technologies based (DNA) base editing, with splice site disruption based genic disruption.

We thank the reviewer for this comment and their careful review.

The core of the paper is formed by an algorithm that picks specific sgRNAs to base-edit splice sites. This algorithm is then used in a head to head comparison of CBE vs ABE editors and point mutation vs splice site deletion efficiencies in knockout generation. The system in which this comparison takes place is the multiplex editing of the T cell immune synapse (a very clever approach since loss of TCR/MHC can be determined by FACS whereas loss of the gene in its various configurations is determined by Sanger sequencing.

We thank the reviewer for their comments.

Running the wet lab experiment was extremely informative (probably beyond what the authors initially intended). Interestingly they found that targeting splice donor sites produced substantially more robust knockouts than pmSTOPs (in itself perhaps not unexpected); that targeting middle exon splice sites was far more efficient than either 5' or 3' eons; and nd targeting splice donor sites was superior to targeting acceptor sites. Perhaps due to the sequence surrounding SD and SA sites, and the unique preferences of each of the base editors, CBEs (rAPOBEC1-BE4) were more reliable than ABEs (ABE7) in generating splice site based protein knockouts.

We thank the reviewer for their appreciation of our work.

Overall this is an excellent paper with robust validation of a proposed tool - but beyond that, with the wet lab providing interesting novel insights that actually lead to optimization of the proposed approach. And beyond: one could argue (e.g fig 5) that they provide inadvertently an excellent window into the mechanism of splicing and into how genic disruption leads to protein loss.

We thank the reviewer for comments and interpretation of our results.

Reviewers' Comments:

Reviewer #1:

Remarks to the Author:

The authors have addressed all of my comments and the manuscript is appropriate for publication in nature communications in its current form in my opinion.

Reviewer #2:

Remarks to the Author:

Authors have responded thoroughly to all my comments